# A multi-factor trafficking site on the spliceosome remodeling enzyme BRR2 recruits C9ORF78 to regulate alternative splicing

Alexandra Bergfort[1,8], Marco Preußner[2], Benno Kuropka [3,4], İbrahim Avşar Ilik[5], Tarek Hilal [1,4,6], Gert Weber [7], Christian Freund [3], Tuğçe Aktaş [5], Florian Heyd[2] & Markus C. Wahl [1,7 ✉]

The intrinsically unstructured C9ORF78 protein was detected in spliceosomes but its role in splicing is presently unclear. We find that C9ORF78 tightly interacts with the spliceosome remodeling factor, BRR2, in vitro. Affinity purification/mass spectrometry and RNA UV-crosslinking analyses identify additional C9ORF78 interactors in spliceosomes. Cryogenic electron microscopy structures reveal how C9ORF78 and the spliceosomal B complex protein, FBP21, wrap around the C-terminal helicase cassette of BRR2 in a mutually exclusive manner. Knock-down of C9ORF78 leads to alternative NAGNAG 3′-splice site usage and exon skipping, the latter dependent on BRR2. Inspection of spliceosome structures shows that C9ORF78 could contact several detected spliceosome interactors when bound to BRR2, including the suggested 3′-splice site regulating helicase, PRPF22. Together, our data establish C9ORF78 as a late-stage splicing regulatory protein that takes advantage of a multi-factor trafficking site on BRR2, providing one explanation for suggested roles of BRR2 during splicing catalysis and alternative splicing.

[1] Freie Universität Berlin, Institute of Chemistry and Biochemistry, Laboratory of Structural Biochemistry, Berlin, Germany. [2] Freie Universität Berlin, Institute of Chemistry and Biochemistry, Laboratory of RNA Biochemistry, Berlin, Germany. [3] Freie Universität Berlin, Institute of Chemistry and Biochemistry, Laboratory of Protein Biochemistry, Berlin, Germany. [4] Freie Universität Berlin, Institute of Chemistry and Biochemistry, Core Facility BioSupraMol, Berlin, Germany. [5] Max Planck Institute for Molecular Genetics, Berlin, Germany. [6] Freie Universität Berlin, Institute of Chemistry and Biochemistry, Research Center of Electron Microscopy and Core Facility BioSupraMol, Berlin, Germany. [7] Helmholtz-Zentrum Berlin für Materialien und Energie, Macromolecular Crystallography, Berlin, Germany. [8] Present address: Yale University, Molecular Biophysics and Biochemistry, New Haven, CT, USA. ✉email: markus.wahl@fu-berlin.de

Splicing of precursor messenger RNAs (pre-mRNAs) is a crucial step in eukaryotic gene expression, and more than 95% of all human multi-exon genes give rise to pre-mRNAs that undergo alternative splicing, thereby massively increasing the genome's coding capacity[1]. The regulation of alternative splicing is highly complex, with many protein and RNA components being involved, and aberrations are often linked to human disease[2]. Splicing is mediated by a complex and dynamic RNA-protein (RNP) molecular machine, the spliceosome. A single round of splicing is characterized by numerous distinct, sequential stages of the spliceosome; each transition between these stages involves profound remodeling of the spliceosome's molecular composition and conformation[3–5]. At least eight conserved RNA helicases/RNP remodeling enzymes drive and regulate key transitions in this splicing cycle, one of which is BRR2 (SNRNP200)[6]. BRR2 is a specific subunit of the U5 small nuclear (sn) RNP that plays a pivotal role during spliceosome activation, the step during which the most substantial rearrangements occur[7,8]. During this transition, BRR2 unwinds the initially base-paired U4/U6 di-snRNA, facilitating the release of U4 snRNA and of all U4/U6 di-snRNP-associated proteins, and allowing U6 to form a catalytically important stem-loop and to engage in alternative interactions with the 5′-splice site (ss) and U2 snRNA[7–9]. However, BRR2 has been suggested to be also involved in the catalytic and disassembly phases of splicing, but the underlying mechanisms are not understood, as these stages do not present known RNA/RNP substrates on which the remodeling factor might act, and as BRR2's ATPase/helicase activity does not seem to be required during these stages[10–12].

BRR2 is structurally and functionally unique among the spliceosomal RNA helicases/RNP remodeling factors. It is the only Ski2-like helicase of the spliceosome and it comprises not a single, but two closely associated and structurally similar helicase units (cassettes)[13]. Only the N-terminal cassette (NC) is an active ATPase and RNA helicase, while the C-terminal cassette (CC) is catalytically inactive, but can regulate the activity of the NC[13,14]. Both cassettes comprise two RecA-like domains, followed by a winged helix (WH) domain and a Sec63 homology module, consisting of helical bundle (HB), helix-loop-helix (HLH), and immunoglobulin-like (IG) domains[13,15,16].

While all other helicases join the spliceosome only transiently, BRR2 is recruited to the pre-catalytic B complex as part of the U4/U6-U5 tri-snRNP and stays associated during all remaining phases of splicing. Furthermore, it encounters its U4/U6 di-snRNA substrate already prior to spliceosome activation within the U4/U6-U5 tri-snRNP, and, consequently, its enzymatic activity needs to be precisely regulated[17]. Apart from several intra-molecular regulatory mechanisms[18–20], the BRR2 activity depends on multiple trans-acting factors[21]. Most importantly, BRR2 is regulated by PRPF8, the largest and most conserved spliceosomal protein that forms a scaffold for the catalytic RNA core of the spliceosome[4,5,22]. PRPF8 encompasses C-terminal RNase H-like and Jab1/MPN-like (Jab1) domains, both of which can regulate BRR2[23–26]. The Jab1 domain can inhibit BRR2 helicase activity by inserting its flexible, C-terminal 16 residues into the BRR2 RNA-binding tunnel; upon removal of the C-terminal Jab1 tail from the tunnel, Jab1 acts as a strong BRR2 activator[21,25].

The largely intrinsically disordered formin-binding protein 21 (FBP21) is one of the non-snRNP accessory proteins, that enters the spliceosomal B complex prior to activation together with eight other B-specific proteins[4,27]. FBP21 and other B-specific proteins are displaced again during the BRR2-dependent conversion of the B complex to the B$^{act}$ complex[4,27]. FBP21 stably interacts with the Sec63 module of the BRR2 CC and inhibits the BRR2-mediated U4/U6 di-snRNA unwinding[28,29]. Two additional, largely intrinsically disordered proteins (IDPs), NTR2 and SNU66, interact with the BRR2 CC in yeast[30–32]. Although the human SNU66 and FBP21 interactions with BRR2 are not entirely visible in recent cryogenic electron microscopy (cryoEM) structures of the human pre-B and B complexes[33–35], there might be a pattern for the BRR2 CC to act as a landing pad for IDPs that can have a splicing regulatory function.

The 34 kDa C9ORF78 protein is predicted to be largely intrinsically disordered. It has been described as a tumor antigen in hepatocellular carcinoma[36] and is considered a marker for a favorable clinical development of renal cancer[37], but its cellular functions are essentially uncharacterized. A C9ORF78 homolog, TLS1, of Schizosaccharomyces pombe has been described as a BRR2 interactor that modulates splicing of shelterin components and thereby affects telomeric heterochromatin assembly and telomere length[38]. Human C9ORF78 has been found associated with spliceosome preparations enriched for the spliceosomal C complex[27], but whether the protein directly interacts with BRR2 and which role it might play in pre-mRNA splicing is presently unknown.

Here, we demonstrate a direct C9ORF78-BRR2 interaction in human and show mutually exclusive binding of C9ORF78 to BRR2 with FBP21. CryoEM-based structural analyses elucidated details underlying this binding competition and allowed targeted disruption of the C9ORF78-BRR2 interaction. System-wide protein and RNA interaction studies revealed additional C9ORF78-interacting components of the spliceosome. SiRNA-mediated knock-down (KD) and complementation in combination with RNA sequencing (RNA-seq) demonstrated a role of C9ORF78 in alternative splicing, in particular with respect to alternative 3′-ss selection and exon skipping, with the latter dependent on the C9ORF78-BRR2 interaction.

## Results

### C9ORF78 interacts with BRR2 in vitro and in HEK293T cells.
We found the human C9ORF78 protein as a putative novel interactor of BRR2 in a large-scale yeast two-hybrid screen. To test whether C9ORF78 binds BRR2 directly, we produced recombinant His$_6$-GST-C9ORF78 (hereafter GST-C9ORF78) in Escherichia coli cells. Tag cleavage led to degradation of the protein and was omitted for functional analyses in vitro. GST-C9ORF78 stably interacted with full-length BRR2 (BRR2$^{FL}$) in analytical size exclusion chromatography (SEC; Fig. 1a). Testing of BRR2 truncation variants revealed that its helicase region (BRR2$^{HR}$, residues 395–2129), encompassing both helicase cassettes but lacking a ~400-residue, auto-regulatory N-terminal region[18], is sufficient for GST-C9ORF78 binding (Supplementary Fig. 1a). However, binding efficiency was reduced for the isolated BRR2 CC (BRR2$^{CC}$, residues 1282–2136), and the BRR2 NC (BRR2$^{NC}$, residues 395–1324) did not co-migrate with GST-C9ORF78 (Supplementary Fig. 1a), suggesting that the interaction takes place predominantly via the CC and is supported by the NC. We also observed co-immunoprecipitation (co-IP) of BRR2 via Flag-C9ORF78 in HEK293T cells (Fig. 1b).

In recent years, cryoEM structures of the human spliceosome revealed persistent BRR2 binding by the PRPF8 Jab1 domain (PRPF8$^{Jab1}$; residues 2064–2335) throughout the splicing cycle[9,33–35,39–41]. We therefore tested if BRR2$^{HR}$ binding by PRPF8$^{Jab1}$ and GST-C9ORF78 can occur simultaneously. Both proteins together co-migrated with BRR2$^{HR}$ in analytical SEC, while no BRR2-independent interaction of PRPF8$^{Jab1}$ and GST-C9ORF78 was detected (Supplementary Fig. 1b).

### CryoEM structure of a BRR2-PRPF8$^{Jab1}$-C9ORF78 complex.
Co-purification of BRR2 and C9ORF78 allowed for His$_6$/GST-tag cleavage without degradation of the C9ORF78 protein. We determined a cryoEM structure of a BRR2$^{HR}$-PRPF8$^{Jab1}$-C9ORF78

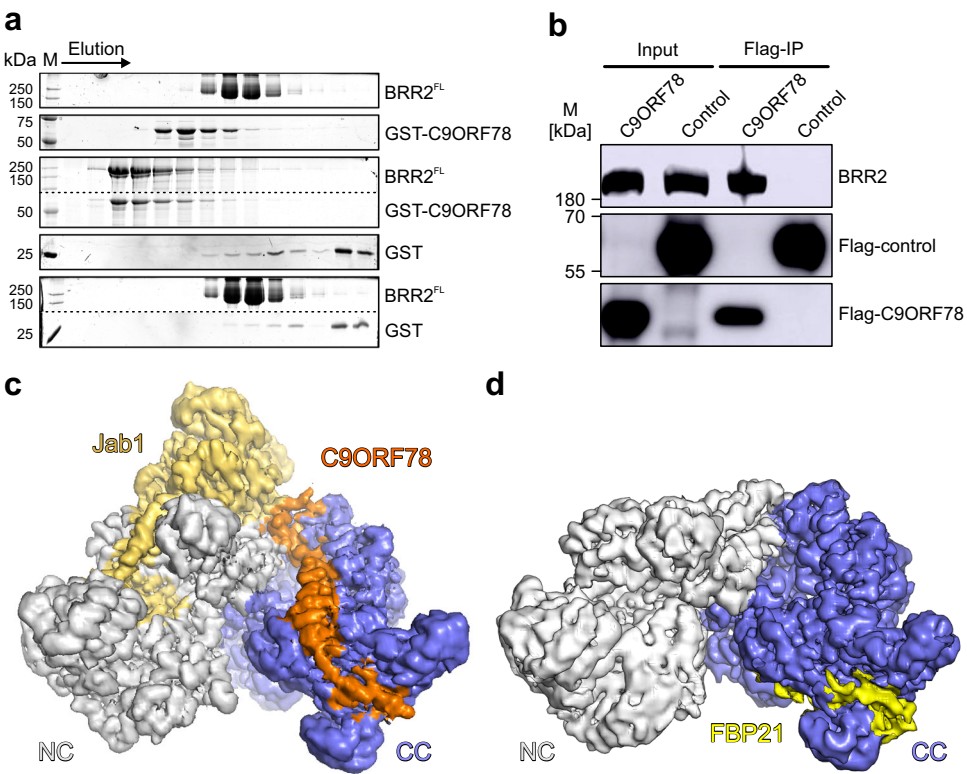

**Fig. 1 C9ORF78 stably interacts with BRR2. a** SDS–PAGE gels showing elution fractions from analytical SEC, monitoring interaction of GST-C9ORF78 with BRR2$^{FL}$. Elution direction is indicated by an arrow. The same elution fractions from runs under identical conditions are shown. Protein bands are identified on the right. M, molecular mass marker. In the third and fifth panel, upper and lower regions of the same gels were spliced together. Dotted lines, splice positions. Independent experiments were conducted at least twice with similar results. **b** C9ORF78 interacts with BRR2 in HEK293T cells. Flag-IP with HEK293T nuclear extract followed by Western blot and antibody staining against BRR2 (top) or Flag-tag (middle and bottom). Flag-CLK1 protein was taken as negative control (Flag-control). Protein bands are identified on the right. M, molecular mass marker. Independent experiments were conducted twice with similar results. Source data for **a** and **b** are provided as a Source Data file. **c** CryoEM map of a BRR2$^{HR}$-PRPF8$^{Jab1}$-C9ORF78 complex at 2.76 Å resolution, contoured at 6 root-mean-square-deviation (RMSD). Color coding in this and the following figures: BRR2 NC, gray; BRR2 CC), slate blue; PRPF8$^{Jab1}$, gold; C9ORF78, orange. **d** CryoEM map of a BRR2$^{HR}$-FBP21$^{200-376}$ complex at 3.3 Å resolution, contoured at 6 RMSD. Color coding in this and the following figures: FBP21$^{200-376}$, yellow.

complex at a nominal resolution of 2.76 Å (Fig. 1c, Supplementary Figs. 2a–e, 3a and 4a, b; Supplementary Table 1). The globular part of PRPF8$^{Jab1}$ resides at its known binding site on the BRR2 NC, with the PRPF8$^{Jab1}$ C-terminal tail penetrating the BRR2$^{HR}$ RNA-binding tunnel (Fig. 1c). C9ORF78 residues 5-58 run along one entire flank of the BRR2 CC in an extended conformation. BRR2 binding by the N-terminal part of C9ORF78 is supported by limited proteolysis and peptide spot array analyses (Supplementary Fig. 1c, d). Upon interaction, C9ORF78 residues 14–18, 20–41, and 49–54 adopt α-helical structures (α1–α3; Fig. 2a). It has been shown that salt bridges formed between oppositely charged residues spaced at positions i and i + 3/i + 4 (ER/K motifs) can stabilize isolated α-helices[42,43], are frequently found in long, single α-helices in spliceosomal proteins and can reduce loss of conformational entropy upon interactions[44]. The long central helix α2 of C9ORF78 exhibits five such motifs, suggesting that it constitutes an important docking device on BRR2.

C9ORF78 residues 5–19 bind to a groove between the HLH and IG domains of the C-terminal Sec63 module. The long central helix α2 runs along the C-terminal HB domain, with its tip contacting the WH domain of the CC. Residues 42–58, including α3, interconnect the CC and NC. Residues 42–48 line a groove between the C-terminal WH and N-terminal IG domains. α3 additionally reinforces the interaction between the globular part of the PRPF8 Jab1 domain and the BRR2 N-terminal IG domain. The four N-terminal and 231

C-terminal residues of C9ORF78 could not be located in the cryoEM map, suggesting that they remain unbound and flexible. The limited contacts of C9ORF78 to the NC and to the PRPF8 Jab1 domain are consistent with these interactions not being stable in isolation, but the NC reinforcing C9ORF78 interaction with the CC (Supplementary Fig. 1a, b).

At the N-terminus of C9ORF78, F8 stacks on F1983 of BRR2 and the following R9 engages in ionic interactions with two glutamic acid residues on BRR2 (E1944 and E2119; Fig. 2a). R41 at the tip of C9ORF78 α2 engages in a particularly large number of polar contacts with BRR2 residues Q1798, S1799, and E1830 (Fig. 3a). In the following linker region, R43 and N45 maintain polar contacts to Q1791 and E1201 of BRR2, respectively. Mixed hydrophobic and hydrophilic contacts ensue in the interaction between the C-terminal 49–58 residues of C9ORF78 and BRR2. F8, R9, R41, R43 and N45 as well as neighboring regions of C9ORF78 are highly conserved (Fig. 2b).

Analysis of the structure with the PISA server[45] revealed a small interaction interface of C9ORF78 with PRPF8$^{Jab1}$ (59 Å$^2$ of interface area), consistent with this interaction not being stable in isolation. In contrast, C9ORF78 exhibits 2,195 Å$^2$ of interface area with BRR2$^{HR}$, comparable to the interface area seen for the BRR2$^{HR}$-PRPF8$^{Jab1}$ interaction (2,698 Å$^2$). Nevertheless, the calculated solvation free energy gain (Δ$^i$G) upon interface formation (reflecting hydrophobic interactions) is more favorable for the BRR2$^{HR}$-PRPF8$^{Jab1}$ compared to the BRR2$^{HR}$-C9ORF78

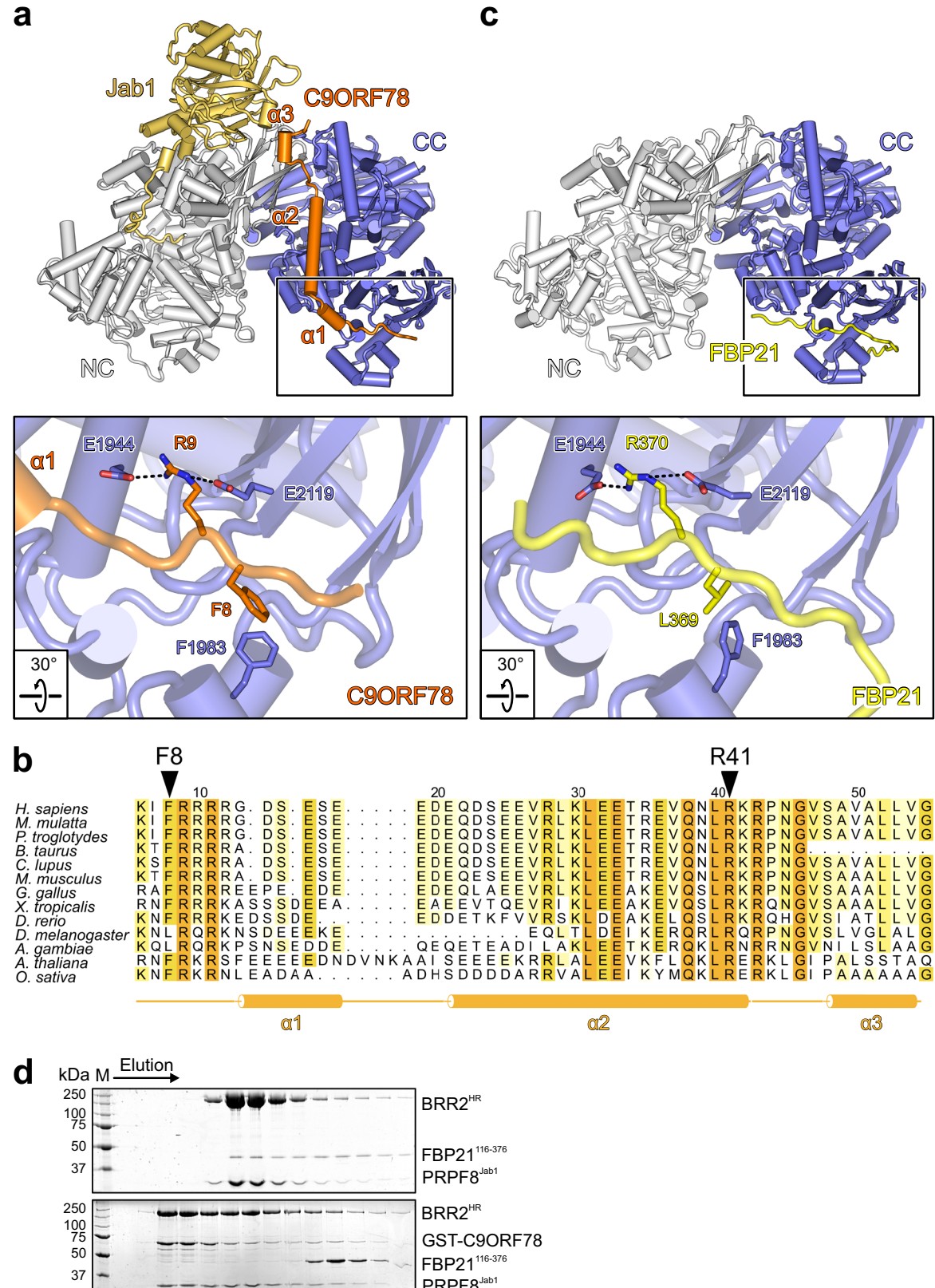

interaction (estimated −16.9 vs. −3.9 kcal/mol). The BRR2$^{HR}$-C9ORF78 interaction relies relatively more on electrostatic interactions; 38 interface residues of C9ORF78 engage in 40 hydrogen bonds and salt bridges to BRR2, contributing an estimated −17.8 kcal/mol to Δ$^i$G, while 58 interface residues of PRPF8$^{Jab1}$ engage in 39 electrostatic contacts to BRR2 (estimated

−17.9 kcal/mol). Therefore, despite similar size interfaces, we expect the BRR2$^{HR}$-C9ORF78 interaction to be significantly weaker and more transient than the BRR2$^{HR}$-PRPF8$^{Jab1}$ interaction, for which a K$_d$ of about 8 nM has been estimated in yeast[46]. Taken together, the mode of BRR2 binding observed for C9ORF78 reflects its intrinsically unstructured nature[47]; it relies

**Fig. 2 BRR2 binding by FBP21 and C9ORF78 is mutually exclusive. a** Cartoon model of the BRR2$^{HR}$-PRPF8$^{Jab1}$-C9ORF78 complex (top) and close-up of the C9ORF78 binding site at the C-terminal Sec63 module of BRR2$^{HR}$ (bottom; boxed in the top image). Selected interacting residues are shown as sticks and labeled. In this and the following structure images: dashed lines, hydrogen bonds or salt bridges. **b** Multiple sequence alignment of C9ORF78 residues 5-58 (BRR2$^{HR}$-binding region) from different species, as indicated on the left. Residues are colored light to dark according to increasing level of conservation (54% conservation, light yellow; 100% conservation, orange). Residue numbering according to human C9ORF78 is indicated above the sequences. Secondary structure elements observed in the BRR2$^{HR}$-PRPF8$^{Jab1}$-C9ORF78 cryoEM structure are indicated below the sequence. Residues R9 (reduced BRR2 binding upon alanine exchange) and R41 (abrogation of BRR2 binding upon alanine exchange) are highlighted by triangles. The alignment was prepared with Homologene (NCBI) employing Clustal Omega[75] and shaded with ALSCRIPT[76]. **c** Cartoon model of the BRR2$^{HR}$-FBP21$^{200-376}$ complex (top) and close-up of the FBP21$^{200-376}$ binding site at the C-terminal Sec63 module of BRR2$^{HR}$ (bottom; boxed in the top image). Selected interacting residues are shown as sticks and labeled. **d** SDS-PAGE gels showing elution fractions from analytical SEC, monitoring competitive binding of FBP21$^{116-376}$ and C9ORF78 to a BRR2$^{HR}$-PRPF8$^{Jab1}$ complex. BRR2$^{HR}$, PRPF8$^{Jab1}$ and FBP21$^{116-376}$ were pre-incubated and complex formation was analyzed without (top) or with (bottom) subsequent addition of GST-C9ORF78. Independent experiments were conducted twice with similar results. Source data are provided as a Source Data file.

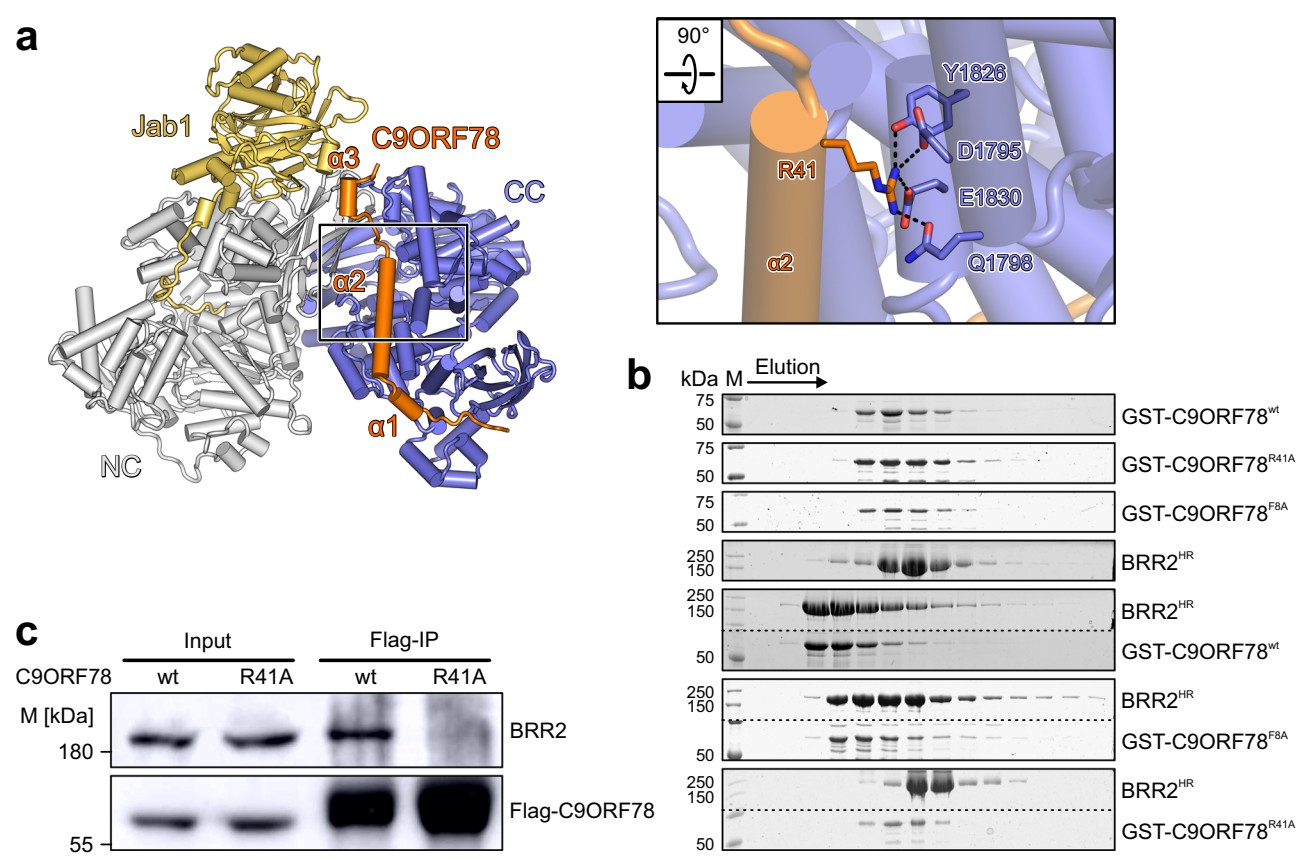

**Fig. 3 Mutational disruption of the C9ORF78-BRR2$^{HR}$ interaction. a** Close-up view of C9ORF78 residue R41 interacting with BRR2 residues Y1826, D1795, E1830 and Q1798. **b** SDS-PAGE gels showing elution fractions from analytical SEC, monitoring interactions of GST-C9ORF78$^{wt}$, GST-C9ORF78$^{F8A}$ and GST-C9ORF78$^{R41A}$ with BRR2$^{HR}$. The same elution fractions from runs under identical conditions are shown. Protein bands are identified on the right. M, molecular mass marker. In the third and fifth panel, upper and lower regions of the same gels were spliced together. Dotted lines, splice positions. Independent experiments were conducted twice with similar results. **c** Flag-IP followed by Western blot with C9ORF78$^{wt}$ or C9ORF78$^{R41A}$. Immunostaining against BRR2 and Flag-tag. M, molecular mass marker. Independent experiments were conducted twice with similar results. Source data for **b** and **c** are provided as a Source Data file.

to a relatively large part on electrostatic interactions; it buries a large interaction surface per residue, likely giving rise to a highly specific interaction; yet it exhibits moderate stability, presumably due to loss of conformational entropy as a consequence of local folding and immobilization upon binding[47].

**C9ORF78 moderately down-regulates BRR2 helicase activity.** The U4/U6 di-snRNA is the only known BRR2 helicase substrate in the spliceosome. Although C9ORF78 has only been detected in spliceosomes after U4/U6 unwinding[27], we employed U4/U6 di-snRNA to test whether C9ORF78 has the potential to affect the BRR2 helicase activity. We performed radioactive, gel-based U4/U6

unwinding assays in the absence or presence of C9ORF78, PRPF8$^{Jab1}$ (BRR2 inhibitor) and/or PRPF8$^{Jab1\Delta C}$ (residues 2064–2319; activator lacking the C-terminal tail). Only a moderate inhibitory effect was observed upon C9ORF78 binding, which was relatively stronger in the presence of PRPF8$^{Jab1}$ or PRPF8$^{Jab1\Delta C}$ (Supplementary Fig. 5). These findings suggest that C9ORF78 may aid in keeping BRR2 in an inactive form during stages of splicing when C9ORF78 is present.

**BRR2 binding by C9ORF78 and FBP21 is mutually exclusive.** The FBP21 C-terminal region is known to interact with the BRR2 CC[28,29]. Therefore, we asked if FBP21 and C9ORF78 occupy a common BRR2 binding site, by determining a

BRR2$^{HR}$-FBP21$^{200-376}$ cryoEM structure at a nominal resolution of 3.3 Å (Fig. 1d, Supplementary Figs. 2f–j, 3b and 4c,d; Supplementary Table 1). The structure revealed that residues 357–375 of FBP21$^{200-376}$ wrap around the C-terminal Sec63 module of BRR2$^{HR}$, as previously suggested by NMR and crosslinking analyses[28,29], with the binding site overlapping a C9ORF78 interaction surface (Fig. 2a, c, top). In particular, FBP21 exhibits an L369-R370 motif that engages the same glutamic acid and phenylalanine residues of BRR2$^{HR}$ (F1983, E1944, E2119) as seen for the F8-R9 motif in C9ORF78 (Fig. 2a, c, bottom).

To test mutually exclusive binding biochemically, we performed SEC analyses with a preformed BRR2$^{HR}$-PRPF8$^{Jab1}$-FBP21$^{116-376}$ complex to which we added GST-C9ORF78 (Fig. 2d). We used a slightly longer FBP21 fragment in these experiments than in cryoEM studies to allow distinction from PRPF8$^{Jab1}$ in SDS-PAGE. While FBP21$^{116-376}$ formed a stable complex with BRR2$^{HR}$-PRPF8$^{Jab1}$ (Fig. 2d, top), GST-C9ORF78 displaced FBP21$^{116-376}$ from the BRR2$^{HR}$-PRPF8$^{Jab1}$-FBP21$^{116-376}$ complex (Fig. 2d, bottom), indicating mutually exclusive binding and a stronger affinity of GST-C9ORF78 for BRR2$^{HR}$-PRPF8$^{Jab1}$. These findings suggest that C9ORF78 may aid in the displacement of FBP21 upon conversion of the B to the B$^{act}$ complex, when FBP21 and other B-specific proteins are released.

**C9ORF78 plays a role in alternative splicing of specific target pre-mRNAs.** To elucidate functions of C9ORF78 during splicing, we conducted an siRNA-mediated C9ORF78 KD in HEK293T cells (KD efficiency of 75–80% after 72 h based on RT-qPCR; Fig. 4a). RNA-seq yielded 35 M reads for the triplicate KD and control samples. rMATS analysis of differential alternative splicing revealed 667 significantly affected alternative splicing events upon C9ORF78 KD, with skipped exons being the most dominant form, followed by mutually exclusive exons and alternative 3′-ss usage (Fig. 4b and Supplementary Data 1). While exon skipping events were up- and down-regulated upon C9ORF78 KD to a similar extent (Fig. 4b, c, left), upstream alternative 3′-ss were predominantly skipped upon C9ORF78 KD, leading to the preferred usage of the downstream alternative 3′-ss (Fig. 4b, c, right).

Further analysis showed that C9ORF78 KD-induced exon skipping is associated with short exons, while exons included upon C9ORF78 KD exhibited an increased average length. Additionally, exons whose inclusion changed upon C9ORF78 KD showed weaker 5′-ss but average-strength 3′-ss, independent of the direction of regulation (Supplementary Fig. 6). For alternative 3′-ss, upstream 3′-ss skipping upon C9ORF78 KD was pronounced when the two alternative 3′-ss were separated by a very small number of nucleotides (nts; median distance of 3 nts between alternative 3′-ss affected upon C9ORF78 KD). Such alternative 3′-ss pairs are referred to as NAGNAG splice sites[48], and most of the C9ORF78-regulated 3′-ss were NAGNAG splice sites (Fig. 4c). In the regulated 3′-ss events, we found the upstream 3′-ss to be generally weaker, indicating a requirement of C9ORF78 to promote usage of weak upstream 3′-ss (Supplementary Fig. 7).

Enhanced skipping of PTBP2 exon 10 and skipping of upstream NAGNAG splice sites in SMARCA4 and C1ORF131 upon C9ORF78 KD were validated via radioactive RT-PCR (Fig. 4d). PTBP2 exon 10 skipping results in a frame shift, generation of a pre-mature termination codon and nonsense-mediated decay[49,50]. As PTBP2 is a splicing factor itself, we compared genes regulated by C9ORF78 with known PTBP2 target genes, as suggested by published crosslinking/IP (CLIP) data[51]. Based on these data, only 26 of the 392 C9ORF78 exon skipping targets are bound by PTBP2. Thus, the majority of exon skipping events affected by C9ORF78

KD cannot be explained by an effect of C9ORF78 on PTBP2 exon 10 skipping. Taken together, our data establish C9ORF78 as an alternative splicing factor that regulates inclusion of certain exons and NAGNAG splice sites.

**C9ORF78 function in alternative splicing at least partially depends on its interaction with BRR2.** To test whether the function of C9ORF78 in alternative splicing is mediated through its interaction with BRR2, we identified interacting residues on BRR2$^{HR}$ and C9ORF78 whose alteration might abolish the interaction. Multiple sequence alignment of C9ORF78 homologs revealed highly conserved residues within regions of C9ORF78 that contact BRR2, including the C9ORF78 F8-R9 motif and the region around R41 in the long central helix of C9ORF78 (Fig. 2b; Supplementary Fig. 8). We exchanged C9ORF78 F8 and R41 individually for alanine residues, and tested BRR2 binding of the C9ORF78 variants via analytical SEC. While C9ORF78$^{F8A}$ showed reduced binding to BRR2, co-migration of BRR2 and C9ORF78$^{R41A}$ was completely abolished (Fig. 3b). Co-IP of BRR2 via Flag-C9ORF78$^{R41A}$ in HEK293T cells suggested that the BRR2-C9ORF78$^{R41A}$ interaction is also reduced in cells (Fig. 3c).

We then transfected HEK293T cells with siRNA-resistant genes encoding either C9ORF78$^{wt}$ or C9ORF78$^{R41A}$, and after 2 h knocked down endogenous C9ORF78 via siRNAs for 72 h. RNA-seq analysis confirmed KD of endogenous C9ORF78 and over-expression of the siRNA-resistant variants to a similar extent (Supplementary Fig. 9). rMATS analysis confirmed the global changes in alternative splicing upon C9ORF78 KD as seen in the first C9ORF78 KD experiment. Significantly changed alternative 3′-ss strongly overlapped between the two KD experiments, with almost all of the overlapping targets being NAGNAG sites (28 of 33; Fig. 5a). Strikingly, we find C98ORF78 KD-induced alternative 3′-ss skipping globally reverted upon both C9ORF78$^{wt}$ and C9ORF78$^{R41A}$ over-expression (Fig. 5b, c), strongly arguing for a C9ORF78-specific effect.

Skipped exon events overlapped to a lower extent between the two KD experiments (Fig. 5d) and we observed only a partial rescue of C9ORF78 KD-induced changes in exon skipping events via the siRNA-resistant variants (Fig. 5e). Nonetheless, 376 exon skipping events were significantly altered in both KD datasets (Δpercent spliced-in [PSI] > 0.1; $p < 0.05$), 49 of which were significantly reverted only by over-production of C9ORF78$^{wt}$ (Fig. 5f), including skipping of PTBP2 exon 10 (Fig. 5g), indicating a regulatory mechanism that depends on the observed BRR2-C9ORF78 interaction. Together, these findings confirm that the observed alternative splicing changes upon C9ORF78 KD are indeed specific and suggest different mechanisms of splicing regulation, as C9ORF78-regulated alternative 3′-ss appear to be less dependent on the BRR2-C9ORF78 interaction than C9ORF78-regulated cassette exons.

**C9ORF78 UV-crosslinks with U5 snRNA.** To test if C9ORF78 regulates alternative splicing through direct interactions with target pre-mRNAs, we generated a Flp-In™ T-REx™ 293 cell line that stably expresses C9ORF78, and performed UV-crosslinking followed by enrichment of C9ORF78-coupled RNAs via Fast Ligation of RNA after some sort of Affinity Purification for High-throughput Sequencing (FLASH)[52]. Green fluorescent protein (GFP)-expressing cells were used as a negative control. RNA-seq yielded 1.8 M reads after filtering PCR duplicates and low quality reads, among which we searched for pre-mRNAs that exhibited C9ORF78-dependent alternative 3′-ss or skipped exons via rMATS analysis using deepTools[53] (Supplementary Fig. 10). No enrichment of specific pre-mRNAs was observed in the C9ORF78 crosslinking experiment compared to the GFP control, arguing

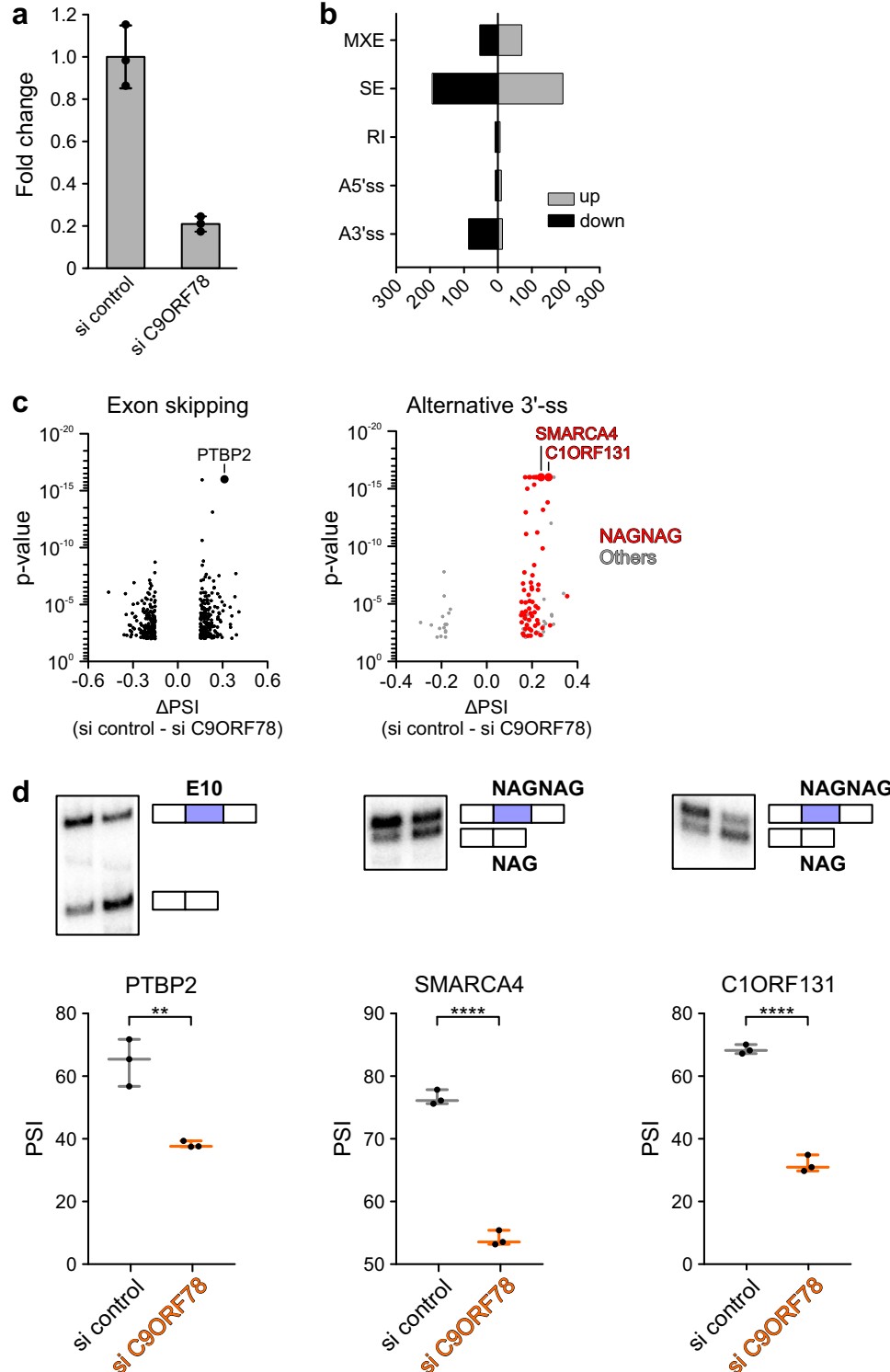

**Fig. 4 C9ORF78 regulates alternative splicing. a** Results from RT-qPCR analyses, showing normalized C9ORF78 expression levels relative to GAPDH expression, in cells transfected with control siRNA (si control) or C9ORF78-targeting siRNA (si C9ORF78). Bars represent means ± SD, $n = 3$ biologically independent experiments. **b** Alternative splicing changes upon siRNA-mediated C9ORF78 KD, as determined by rMATS. MXE, mutually exclusive exons; SE, skipped exon; RI, retained intron; A5′/3′ss, alternative 5′/3′-splice sites. More inclusion upon C9ORF78 KD is indicated in light gray (up), skipping in black (down). **c** Volcano plots of significantly changed skipped exons (left; $n = 390$), and alternative 3′-splice sites (right, $n = 105$; red, NAGNAG splice sites, $n = 59$) upon C9ORF78 KD. Targets selected for validation PCR (PTBP2, C9ORF131, SMARCA4) are indicated by large data points and labeled. **d** Validation PCRs confirm C9ORF78 KD-induced changes in alternative splicing. Top, representative gels. Bottom, quantifications ($n = 3$ biologically independent experiments). Horizontal lines, medians; whiskers, minimum/maximum values. PSI, percent spliced-in (gel analysis), ratio of the quantified band representing exon inclusion and the sum of the quantified bands representing exon inclusion and exon skipping. Statistical significance was determined by unpaired, two-sided t-tests; \*\*$p = 0.0038$; \*\*\*\*$p < 0.0001$. Source data for **a**–**d** are provided as a Source Data file.

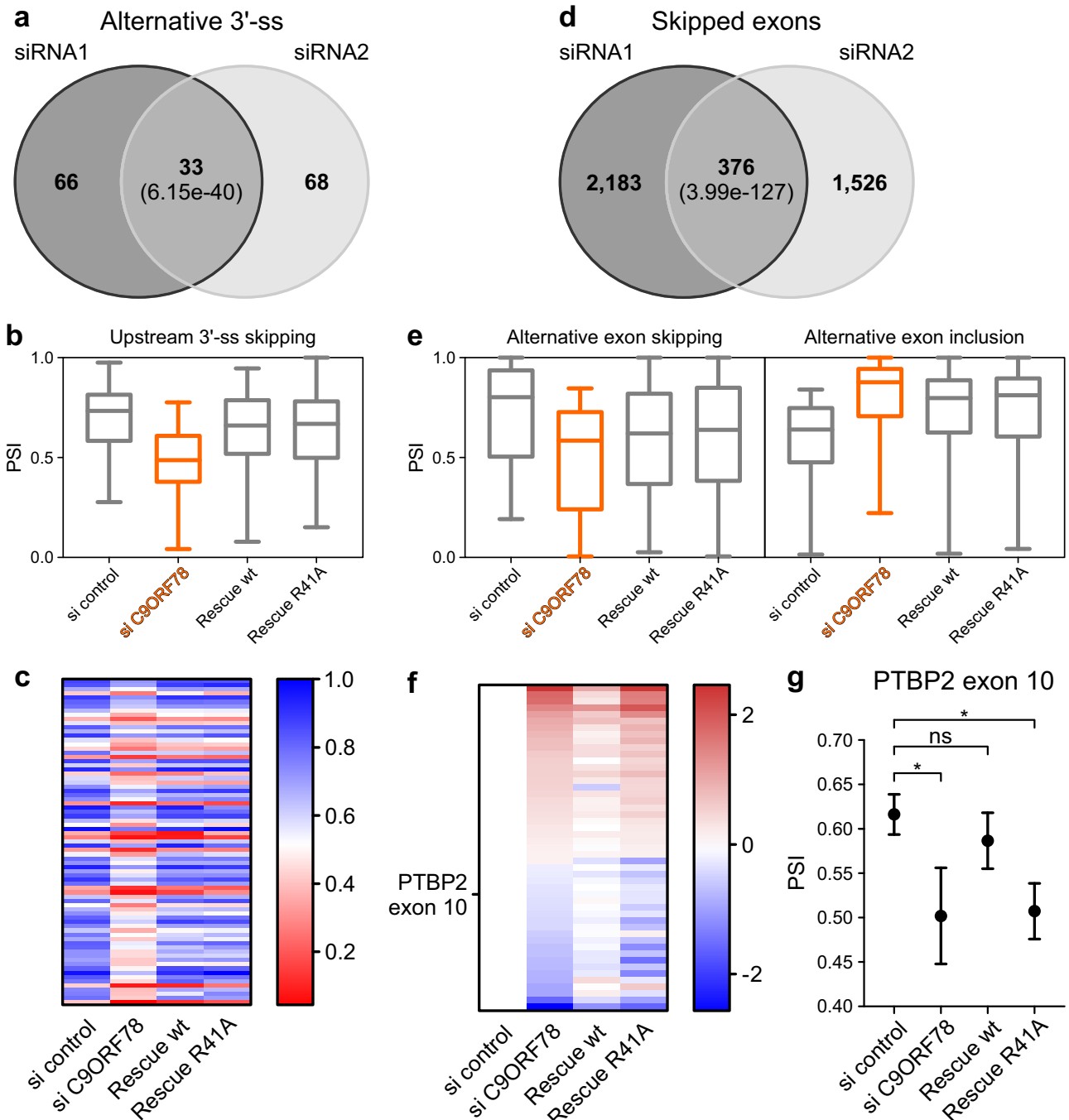

against regulatory mechanisms that rely on direct C9ORF78-pre-mRNA interactions. However, snakePipes[54] analysis revealed a significant enrichment of U5 snRNA (log2-fold change 2.25; p-value 4.290826e-31) and L13AP, a retrotransposon of the LINE-1 family (log2-fold change 0.59; p-value 5.475290e-11) among C9ORF78-crosslinked RNAs (Fig. 6a). A cluster of gaps in the U5 snRNA sequencing reads suggested crosslinks of C9ORF78 to U5 snRNA residues 69–73, which form internal loop 1 (IL1) at the base of the extended U5 5'-stem-loop (Fig. 6b). The suggested direct C9ORF78-U5 snRNA contact supports the notion that C9ORF78 functions in alternative splicing depend on interactions with U5 snRNP components.

**C9ORF78 interacts with additional spliceosomal proteins**. To identify possible additional spliceosomal protein interactors of

C9ORF78, we generated Flp-In™ T-REx™ 293 cell lines stably expressing Flag-C9ORF78[wt] or Flag-C9ORF78[R41A], and conducted Flag-IPs with nuclear extracts, followed by mass spectrometric identification of the recovered proteins. Enrichment was calculated in comparison to a control IP (cells not expressing a Flag-C9ORF78 variant). In total, we quantified 2,411 IP-enriched proteins (Supplementary Data 2). Filtering for a two-fold enrichment over the control (log2-fold change > 1) and a significant t-test result (q < 0.05) yielded 560 enriched proteins in the C9ORF78[wt] Flag-IP and 809 enriched proteins in the C9ORF78[R41A] Flag-IP. We subjected proteins with a log2-fold enrichment > 3 in either of the Flag-IPs to a GO analysis, which indicated U5 snRNP as the most enriched GO term (Fig. 7a).

Spliceosomal proteins enriched in the IPs are shown in Fig. 7b. While C9ORF78[wt] and C9ORF78[R41A] were enriched at almost

**Fig. 5 Rescue of C9ORF78 KD-induced splicing changes. a** Comparison of C9ORF78 KD-induced 3′-ss changes detected in C9ORF78 KD experiments 1 (siRNA1, left) and 2 (siRNA2, right; ΔPSI > 0.15; $p < 0.01$). Numbers in each circle, affected events observed in the separate experiments. Number in the center, events observed in both experiments. Numbers in parentheses, significance of the observed overlaps determined by hypergeometric test. PSI, percent spliced-in (RNA-seq analysis), ratio of quantified junction reads representing exon inclusion to sum of quantified junction reads representing exon inclusion and exon skipping. **b** Box-whiskers plots for C9ORF78-induced 3′-ss skipping events and rescue by siRNA-resistant C9ORF78 variants (based on second KD experiment including rescue; ΔPSI > 0.15; $p < 0.01$; $n = 77$ regulated alternative 3′-ss). Horizontal lines, medians; whiskers, minimum and maximum values; boxes, upper and lower quartile. **c** Rescue of alternative 3′-ss events by siRNA-resistant C9ORF78 variants (based on the second KD experiment that included rescue). Heat map, mean PSI values for 3′-ss events significantly (ΔPSI > 0.15; $p < 0.01$) and reproducibly altered upon C9ORF78 KD, showing that 3′-ss changes are reverted similarly by C9ORF78[wt] and C9ORF78[R41A]. **d** Comparison of C9ORF78 KD-induced changes in exon skipping detected in C9ORF78 KD experiments 1 (siRNA1, left) and 2 (siRNA2, right; ΔPSI > 0.1; $p < 0.05$). Numbers as in **a**. **e** Box-whiskers plots for C9ORF78-induced cassette exon events (exon skipping, left, $n = 205$; exon inclusion, right, $n = 271$) and rescue by siRNA-resistant C9ORF78 variants (based on second KD experiment including rescue; ΔPSI > 0.15; $p < 0.01$). Symbols as in **b**. **f** Rescue of alternative exon events by siRNA-resistant C9ORF78 variants. Heat map, alternative exon events (log2-fold-change relative to control) significantly (ΔPSI > 0.1; $p < 0.05$) and reproducibly altered upon C9ORF78 KD, showing that these events are specifically reverted by C9ORF78[wt] (ΔPSI > 0.1; $p < 0.05$) but not by C9ORF78[R41A] (ΔPSI < 0.01; $p > 0.01$; $n = 49$ target exons). PTBP2 exon 10, highlighted on the left. **g** Effect of C9ORF78 KD on PTBP2 exon 10 splicing and differential rescue by siRNA-resistant C9ORF78 variants. PSI values (mean ± SEM) were determined by rMATS. Statistical significance is based on rMATS-derived p-values (*$p < 0.05$; ns, not significant; si control vs. si C9ORF78, $p = 0.03$; si control vs. wt, $p = 0.47$; si control vs. rescue R41A, $p = 0.01$; $n = 3$ biologically independent experiments). Source data for **b**, **d** and **g** are provided as a Source Data file.

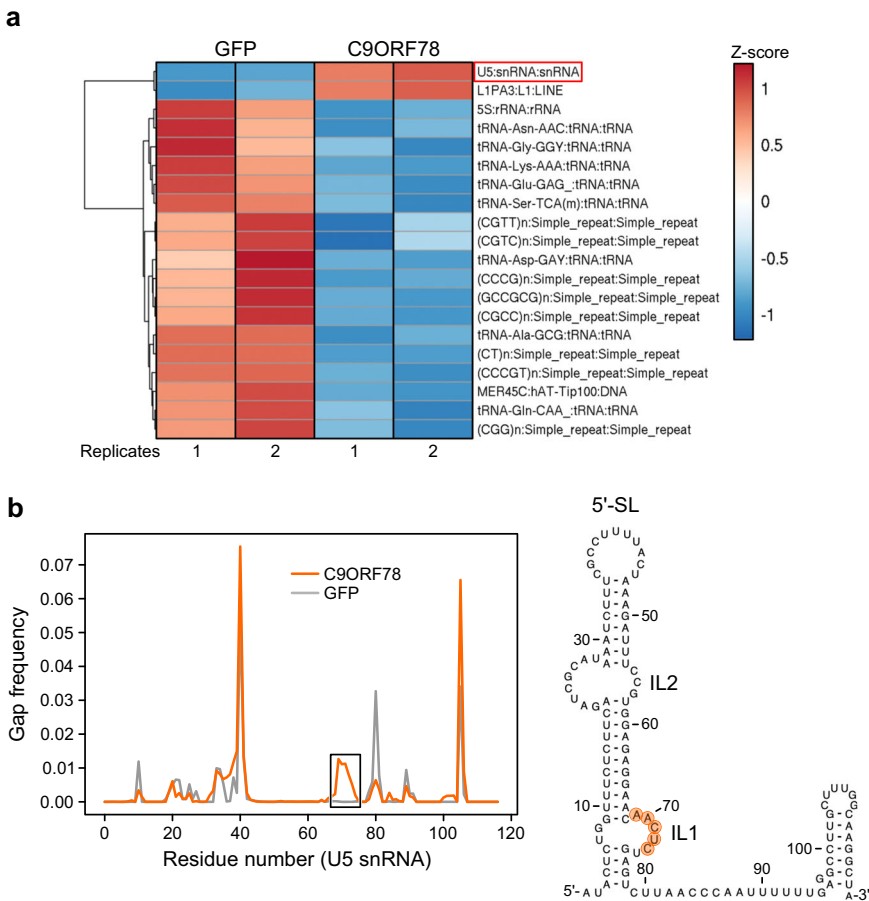

**Fig. 6 Interaction of C9ORF78 with U5 snRNA. a** Heat map displaying the top 20 differentially enriched non-coding RNAs in GFP (negative control) and C9ORF78 pull-down after RNA-protein UV-crosslinking and FLASH[52], analyzed with snakePipes[54]; $n = 2$ biologically independent experiments each are shown. Colored by Z-score as indicated. Negatively enrichment RNAs (blue) in the C9ORF78 pull-down are mainly tRNAs, which are commonly found as background in such analyses. **b** Left, gaps in sequencing reads for U5 snRNA from FLASH experiments with C9ORF78 (orange) or GFP (control; gray). Right, putative C9ORF78 crosslinking sites (orange background) mapped to a secondary structure model of U5 snRNA. 5′-SL, 5′-stem-loop; IL1, internal loop 1, IL2, internal loop 2.

equal levels, enrichment of the U5 snRNP proteins BRR2, PRPF8, EFTUD2, SNRNP40, and PRPF6, the intron-binding complex (IBC) protein SYF1, the B complex protein RED, the B[act] proteins CWC27 and CWC22, the 1st step factor GPKOW and the C* complex protein DHX35 was strongly reduced or abrogated in the Flag-C9ORF78[R41A] IP compared to the Flag-C9ORF78[wt] IP. Reduced enrichment of BRR2 by C9ORF78[R41A] confirmed the weakening of the direct BRR2 interaction via an R41A exchange in C9ORF78. However, the burial of R41 in the C9ORF78-BRR2[HR] interface (Fig. 3a) precludes a direct effect of the R41A

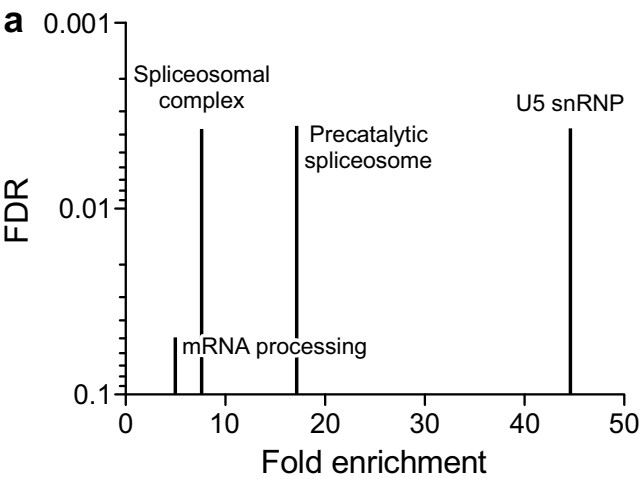

**Fig. 7 Proteins enriched in Flag-C9ORF78[wt] and Flag-C9ORF78[R41A] IPs. a** GO term analysis for proteins significantly enriched in C9ORF78 IPs (log2-fold change >3). Genes expressed in HEK293T cells were used as a background list. Fold enrichment is depicted on the x-axis, the false discovery rate (FDR) on the y-axis. **b** Spliceosomal proteins enriched in Flag-IP from nuclear extract of Flp-In™ T-REx™ 293 cells stably expressing Flag-C9ORF78[wt] (orange) or Flag-C9ORF78[R41A] (light orange). Proteins were assigned to complexes/stages according to [4]. Other co-precipitated proteins are listed in Supplementary Data 2. Data represent log2-fold protein intensity differences of Flag-C9ORF78 IP (wt or R41A) vs. control Flag-IP, $n = 3$ technical replicates. Source data are provided as a Source Data file.

exchange on the enrichment of the other proteins. While direct contacts of C9ORF78 to these proteins through other C9ORF78 regions are certainly possible, reduced enrichment of the proteins in the Flag-C9ORF78$^{R41A}$ IP suggest that such putative binary interactions would depend on concomitant stable interaction of C9ORF78 with BRR2. On the other hand, the intron–binding complex (IBC) protein, aquarius (AQR), and the second step factor, PRPF22 (DHX8), were enriched at comparable levels in both IPs (a slight reduction in the Flag-C9ORF78$^{R41A}$ IP is most likely due to a slight difference in the enrichment of Flag-C9ORF78$^{wt}$ and Flag-C9ORF78$^{R41A}$ themselves). Thus, C9ORF78 may contact AQR and PFPF22 within the spliceosome directly or indirectly, but largely independent of its interaction with BRR2. Finally, we also observed enhanced or exclusive enrichment of PRPF28/DDX23 (U5), PRPF38A (B), Cactin (C*) and MAGOH and Y14 (exon junction complex) in the Flag-C9ORF78$^{R41A}$ IP (Fig. 7b). While the basis for these latter effects is presently unclear, our Flag-IPs again suggest that C9ORF78 might associate with the spliceosome already during the B-to-B$^{act}$ transition and that it may remain bound until the second step.

## Discussion

Structural analysis revealed how C9ORF78 associates with a BRR2$^{HR}$-PRPF8$^{Jab1}$ complex predominantly via the BRR2 CC. The BRR2 CC has been suggested to serve as a binding platform for other spliceosomal proteins that could regulate the BRR2 helicase activity. E.g., NMR and crosslinking studies, validated herein, suggested that a C-terminal, unstructured region of FBP21 wraps around the C-terminal Sec63 module of BRR2[28,29], eliciting a strong inhibitory effect of BRR2 helicase activity[28], consistent with a suggested role for FBP21 in preventing BRR2 from pre-maturely unwinding U4/U6[33]. The molecular basis for this effect is presently not clear but may involve modulation of CC flexibility or NC-CC contacts. The BRR2-modulatory activity of C9ORF78 we report here is weaker, and C9ORF78 seems to be associated with the spliceosome only at stages when BRR2 has already unwound U4/U6. However, we presently cannot exclude that C9ORF78-dependent regulation of BRR2 helicase activity may play a role during other stages of splicing. C9ORF78 and FBP21 occupy a common binding site on BRR2 CC, using similar interaction motifs. This common binding mode to BRR2 explains the mutually exclusive binding of C9ORF78 and FBP21 to BRR2 we observe. Based on these findings, we suggest that the FBP21/C9ORF78 binding competition on a multi-factor trafficking site of BRR2 represents an important principle in the spliceosome, which facilitates ordered, stage-specific recruitment of splicing factors.

The mutually exclusive binding of FBP21 and C9ORF78 to BRR2 suggests that C9ORF78 might first bind to the spliceosome during the B-to-B$^{act}$ transition, when FBP21 is released. While proteomics analyses have suggested that C9ORF78 might be associated with the C complex[27], the analyzed complexes had been enriched on a modified pre-mRNA lacking a 3′-ss AG dinucleotide and a 3′-exon[55]. CryoEM and biochemical studies have shown that spliceosomes assembled on such modified pre-mRNAs can progress to the C* complex state[40], that neighboring states tend to converge on the C complex state when exon ligation is inhibited[56] and that under appropriate conditions both catalytic steps of splicing can be reversed[57]. Thus, factors identified via proteomics in nominal C complex preparations may to some extent represent contaminations from neighboring states. Presence of C9ORF78 already during the B$^{act}$ stage is further supported by putative interactions we observe with the B$^{act}$ proteins CWC22 and CWC27. However, as the C9ORF78-binding site of BRR2 remains unobstructed in C, C* and P complexes[34,40,41,58,59], and as we also

identified putative C9ORF78 interactions with 1$^{st}$ step, C* and 2$^{nd}$ step factors, C9ORF78 may also remain bound after the B-to-B$^{act}$ transition.

The additional interactions of C9ORF78 suggested by our pull-down experiments are consistent with its intrinsically unstructured nature. The 231 C-terminal residues of C9ORF78 are not visible in our BRR2$^{HR}$-PRPF8$^{Jab1}$-C9ORF78 cryoEM structure (Fig. 1c). Many of these residues are highly conserved, pointing towards a functional importance (Supplementary Fig. 8). Obviously, this region could harbor binding sites for other spliceosome components, such as the proteins enriched in our pull-down experiments or U5 snRNA IL1 identified in our FLASH analysis. The C9ORF78 C-terminal region could extend far away from the N-terminal anchor on BRR2 and reach distal regions of the spliceosome. Furthermore, one can expect interactions to be maintained through short C9ORF78 epitopes, as is frequently observed in IDPs[47], such that C9ORF78 could engage in direct contacts to multiple additional spliceosome components at the same time.

C9ORF78 KD elicited changes in many exon skipping events, a significant number of which are dependent on the BRR2–C9ORF78 interaction. Exon skipping is thought to be decided before the C complex stage, providing additional indirect evidence that C9ORF78 is already present at an earlier stage. We also observed a highly reproducible effect of C9ORF78 KD on alternative usage of NAGNAG 3′-ss, with C9ORF78 strongly favoring usage of the upstream 3′-ss, for which differential recue experiments indicated C9ORF78 specificity but failed to support a dependence on the observed BRR2-C9ORF78 interaction. While these data suggest that different C9ORF78-splicing factor interactions play a predominant role for the regulation of cassette exons and of alternative 3′-ss, assay limitations may also have prevented the detection of subtle effects of the BRR2–C9ORF78 interaction on some alternative splicing events. E.g., the single residue exchange in C9ORF78$^{R41A}$ is sufficient to destabilize the binary interaction with BRR2 in vitro, but C9ORF78 interaction with other spliceosomal factors and overexpression of the siRNA-resistant C9ORF78 variants may have obscured differences in rescue efficiencies between C9ORF78$^{wt}$ and C9ORF78$^{R41A}$.

Exon skipping might be influenced by the kinetics with which two mutually exclusive splicing scenarios transition from the B via the B$^{act}$ to the B* stage, and our findings suggest that C9ORF78 could modulate these transitions. Recently, additional assembly intermediates between the B and B$^{act}$ stages have been characterized biochemically and structurally[60]. These pre-B$^{act}$ complexes contain, among others, reduced levels of the B-specific FBP21 protein, but also largely lack B$^{act}$ proteins CWC22 and CWC27 and the step 1 factor GPKOW. Given our observations that C9ORF78 can displace FBP21 from BRR2 and could also contact CWC22, CWC27 and GPKOW, the presence of C9ORF78 might modulate the kinetics of B-to-B$^{act}$ conversion by driving displacement of FBP21 and helping recruitment of B$^{act}$ proteins and GPKOW. Notably, the multi-step B-to-B$^{act}$ transition is also accompanied by a stepwise repositioning of BRR2[60], which might likewise be influenced by C9ORF78 that putatively links BRR2 to other components according to our data. Moreover, a large-scale cryoEM analysis has revealed that the human B$^{act}$ complex can adopt at least eight major conformations, which could be arranged along a trajectory towards catalytic activation due to the degree of their similarity to a later intermediate[61]. Such a situation most likely also applies to other splicing stages, and it has been suggested that any additional incoming factor will alter the conformational space available to the respective spliceosomal intermediate[61]. Based on a structural superposition, it is easily conceivable that in B$^{act}$ the intrinsically unstructured C9ORF78 could bridge between BRR2, CWC22/CWC27 and U5 IL1 (Fig. 8a) This presumed cross-strutting of several B$^{act}$ elements would most

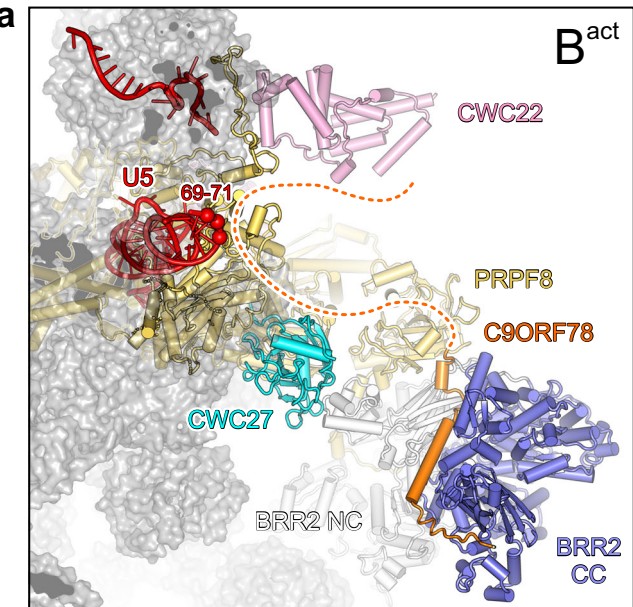

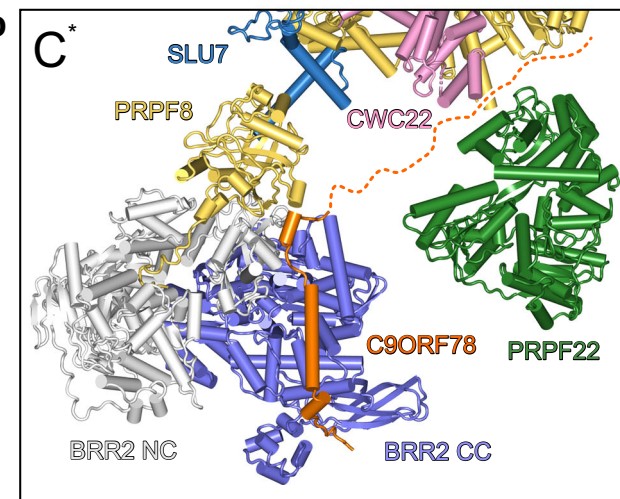

**Fig. 8 Putative C8ORF78 neighborhoods in spliceosomal complexes. a** Model for the positioning of C9ORF78 in the B^act complex, suggesting how C9ORF78 may contact its putative interactors U5 snRNA IL1 (red spheres), CWC22 (cyan), and CWC27 (pink). The BRR2^HR-PRPF8^Jab1-C9ORF78 structure was aligned with a cryoEM structure of the spliceosomal B^act complex (PDB ID 5Z56) according to the common parts of BRR2^HR/BRR2. The 231 C-terminal C9ORF78 residues that are not visible in the BRR2^HR-PRPF8^Jab1-C9ORF78 cryoEM map are indicated by a dashed line. **b** Model for the positioning of C9ORF78 in the C* complex, suggesting how C9ORF78 may contact its putative interactors PRPF22 (green) and CWC22 (pink). SLU7, sky blue. The BRR2^HR-PRPF8^Jab1-C9ORF78 structure was aligned with a cryoEM structure of the spliceosomal C* complex (PDB ID 5XJC) according to the common parts of BRR2^HR/BRR2. The 231 C-terminal C9ORF78 residues that are not visible in the BRR2^HR-PRPF8^Jab1-C9ORF78 cryoEM map are indicated by a dotted line.

likely significantly alter the conformational space available to B^act. C9ORF78 might thereby again alter the kinetics of B-to-B^act conversion and/or influence the tendency of the B^act complex to adopt a conformation conducive to PRPF2 remodeling.

Alternative NAGNAG splice site choice was suggested to take place during the second step of the splicing reaction[48], indirectly supporting our notion of a continued presence of C9ORF78 at post-B^act stages. Furthermore, our observed interaction of

C9ORF78 with the second step factor, PRPF22, suggests that C9ORF78 remains associated also with the C* complex. Comparison of our BRR2^HR-PRPF8^Jab1-C9ORF78 structure with the structure of a human C* complex revealed that the C-terminal 231 residues of C9ORF78 could easily reach and directly contact PRPF22, which resides in the immediate vicinity of BRR2 in the C* complex, as well as CWC22 that is still present at the C* stage (Fig. 8b). PRPF22 has been shown to be involved in 3′-ss selection and exon ligation in yeast[62,63]. Thus, one possible mechanism for C9ORF78 to regulate alternative 3′-ss usage may be direct C9ORF78-PRPF22 interactions that affect PRPF22 motor activity, which is thought to reposition alternative 3′-ss in the spliceosome's active site from a distance[4,63]. This interpretation is consistent with our observation that the presence of C9ORF78 leads to preferential use of upstream alternative 3′-ss.

Previous studies suggested that alternative splicing decisions are not necessarily fixed during initial exon/intron definition[64]. Our findings provide additional examples of alternative splicing regulation comparatively late in the splicing cycle. In addition, our observations point towards hitherto under-appreciated, additional functions of BRR2 during the catalytic phase of splicing, and to novel functions of BRR2 in the regulation of alternative splicing. While requirements for BRR2, beyond U4/U6 unwinding, during the catalytic and disassembly stages of splicing have been suggested, at least in yeast[10,11], the mechanistic basis for these observations is not clear, as BRR2 does not seem to resort to its ATPase/helicase activities to exert these additional roles[10,12]. Our work provides a blueprint for how BRR2 might exert ATPase/helicase-independent functions during splicing, i.e., by serving as a home base for other splicing factors, including stage-specifically recruited splicing regulators. This model also suggests how BRR2, a constitutive splicing factor, can play a role in alternative splicing by recruiting a hitherto poorly characterized, splicing regulatory IDP, C9ORF78.

## Methods

**Protein production.** BRR2 and PRPF8 constructs were produced and purified as described before[13,25]. C9ORF78 variants were produced as His6-GST-tagged proteins from pETM30 plasmids in a BL21 RIL *E. coli* strain. The cell pellet was resuspended in lysis buffer (20 mM Tris-HCl, pH 7.5, 200 mM NaCl, 1 mM DTT), supplemented with protease inhibitors (Roche) and DNaseI, and lysed by sonication. The GST-tagged protein was subjected to affinity chromatography using Glutathione Sepharose 4 Fast Flow resin (GE Healthcare). To reduce chaperone contamination, a wash with 50 mM Tris-HCl, pH 7.5, 10 mM ATP, 10 mM MgCl2, 150 mM KCl before elution. The protein was further purified by SEC on a 16/60 Superdex 200 column (GE Healthcare), equilibrated in 20 mM Tris-HCl, pH 7.5, 200 mM NaCl, 2 mM DTT.

For reconstitution of the BRR2^HR-C9ORF78 complex, lysates of the individually expressed proteins (His10-BRR2^HR in insect cells and His6-GST-C9ORF78 in *E. coli*) were mixed after sonication. Complexes were subjected to affinity chromatography using Glutathione Sepharose 4 Fast Flow resin as described above, followed by overnight dialysis against 20 mM Tris-HCl, pH 7.5, 200 mM NaCl, 2 mM DTT, and TEV protease at a 1:20 ratio. Cleaved proteins were recycled on Ni^2+-NTA Sepharose (GE Healthcare) and further purified by gel filtration on a 16/60 Superdex 200 column equilibrated in 20 mM Tris-HCl, pH 7.5, 200 mM NaCl, 1 mM DTT.

For cryoEM, 1.8 nmol of the BRR2^HR-C9ORF78 complex were incubated with a 1.5-fold molar excess of PRPF8^Jab1 for 10 min on ice. Subsequently the complex was purified by size exclusion chromatography on a Superdex 200 increase 3.2/300 column (GE Healthcare), equilibrated in 20 mM Tris-HCl, pH 8.0, 150 mM NaCl, 1 mM DTT, and then concentrated to 5 mg/ml using ultrafiltration (Amicon, Merck Chemicals GmbH).

FBP21 was purified as described before[28]. For cryoEM 1.8 nmol BRR2^HR were incubated with a two-fold molar excess of FBP21^200–376 for 10 min on ice. Subsequently, the complex was purified as described above for BRR2^HR-PRPF8^Jab1-C9ORF78.

**Analytical size exclusion chromatography.** Analytical SEC was performed on a Superdex 200 increase 3.2/300 column, equilibrated in 20 mM Tris-HCl, pH 8.0, 200 mM NaCl, and 1 mM DTT. Proteins or protein mixtures were pre-incubated on ice for 10 min in 50 µl reaction volumes, containing 75 µg of the respective individual protein, or 75 µg of BRR2 construct or PRPF8^Jab1 and a 1.5-fold molar

excess of putative interaction partner, prior to injection on the column. A total of 60 μL elution fractions were collected and analyzed by SDS-PAGE.

For testing mutually exclusive binding of C9ORF78 and FBP21[116–376] to BRR2[HR]-PRPF8[Jab1], 75 μg BRR2[HR] were incubated with equimolar amounts of PRPF8[Jab1] and FBP21[116–376] for 10 min on ice. Subsequently, equimolar amounts GST-C9ORF78 were added and the protein mixture was incubated for an additional 10 min on ice. Complex formation was analyzed via analytical SEC as described above in comparison to a sample without added GST-C9ORF78.

**GST pull-down assay.** Glutathione Sepharose 4 Fast Flow resin was equilibrated in binding buffer (20 mM Tris-HCl, pH 8.0, 150 mM NaCl, 1 mM DTT). For each of 5 reactions, 20 μl resin slurry were transferred to 1.5 ml Eppendorf tubes and incubated with 50 μg GST-C9ORF78 and equal molar amounts of BRR2[HR] in a total volume of 500 μl (dilution in binding buffer) for 1 h at 4 °C on a rotating wheel. Subsequently, the resin was washed three times with 1 ml binding buffer (centrifugation at 500 g for 3 min) and then incubated with a 0-10-fold molar excess of FBP21[200–376] in 50 μl binding buffer for 20 min at 4 °C on a shaker (600 rpm). The supernatants were transferred to new tubes and kept on ice, while the resin was washed three times with 1 ml binding buffer. Elution from the beads was carried out by incubating the resin in 50 μl SDS-PAGE loading dye at 96 °C for 5 min, followed by 2-min centrifugation at maximum speed in a table-top centrifuge. A total of 10 μl elution (beads) and supernatant were used for SDS-PAGE analysis.

**Limited proteolysis of BRR2[HR]-C9ORF78.** A total of 26 μM complex were digested with 0.052 μg elastase in a 50 μl reaction in 50 mM Tris-HCl, pH 8.0, 200 mM NaCl, 10% (v/v) glycerol, 1 mM DTT for 45 min at room temperature. Subsequently, the reaction was loaded on a Superdex increase 3.2/300 column, equilibrated in the same buffer. A total of 60 μl elution fractions were collected, of which 15 μl were analyzed via SDS-PAGE. Bands of interest were excised, in-gel digested with trypsin and analyzed by mass spectrometry.

**Peptide SPOT array.** Membranes with spots of peptides (25 residues, overlap of 20 residues) of C9ORF78 were obtained from Dr. Rudolf Volkmer, Charité—Universitätsmedizin Berlin. Membranes were pre-washed once with 100% ethanol and three times with phosphate-buffered saline (PBS), supplemented with 1 mM DTT. The remaining binding capacity of the membranes was blocked by a 3-h incubation with blocking buffer (5% (w/v) BSA in PBS, supplemented with 1 mM DTT). Subsequently, the membranes were incubated overnight at 4 °C with His[10]-BRR2[NC], His[10]-BRR2[CC], His[10]-BRR2[HR] or His[10]-BRR2[FL] at a concentration of 25 μg/ml (His[10]-BRR2[NC], His[10]-BRR2[CC]), 50 μg/ml (His[10]-BRR2[HR]) or 60 μg/ml (His[10]-BRR2[FL]) in binding buffer (10 mM Tris-HCl, pH 7.5, 200 mM NaCl, 2 mM DTT). As a negative control, one membrane was incubated with binding buffer without added protein. The membranes were then washed three times with PBS, supplemented with 0.05% (v/v) Tween 20, 1 mM DTT (PBST) and incubated with an HRP-coupled anti-His antibody (Miltenyi Biotech) diluted 1:5000 in PBS with 5% (w/v) BSA for 1 h at room temperature. After washing the membranes three times in PBST, the peptide SPOT arrays were developed with HRP juice (p.j.k. GmbH) on an Intas Advanced Fluorescence and ECL imager.

**U4/U6 snRNA unwinding assays.** Yeast U4 and U6 snRNA production (commonly used also in unwinding assays with human BRR2), purification, labeling and assembly were carried out as described before[65]. All U4/U6 unwinding assays were performed at 30 °C. Unwinding of 0.6–1.5 nM radioactive U4/U6 di-snRNA was compared for 100 nM BRR2 (FL or HR) alone with 100 nM BRR2 (FL or HR) in complex with 150 nM GST-C9ORF78 in the presence or absence of 250 nM PRPF8[Jab1] or PRPF8[Jab1ΔC]. All reactions were pre-incubated for 3 min in unwinding buffer (40 mM Tris-HCl, pH 7.5, 50 mM NaCl, 8% (v/v) glycerol, 0.5 mM MgCl₂, 15 ng/μl acetylated BSA, 1 U/μl RNase inhibitor, 1.5 mM DTT) in a total volume of 120 μl. Unwinding was initiated by addition of 1.7 mM ATP/MgCl₂, and 10 μl samples were withdrawn at the indicated time points and mixed with 10 μl of stop buffer (40 mM Tris-HCl, pH 7.4, 50 mM NaCl, 25 mM EDTA, 1% (w/v) SDS, 10% (w/v) glycerol, 0.05% (w/v) xylene cyanol, 0.05% (w/v) bromophenol blue). The samples were run on a 6% non-denaturing PAGE gel for 1 h at 200 V and 4 °C. RNA bands were visualized by autoradiography using a phosphoimager and quantified using the Image Quant 5.2 software (GE Healthcare). Data were fit to a single exponential equation (fraction unwound = A (1–exp[- $k_u$t])); A, amplitude of the reaction; $k_u$, apparent first-order rate constant of unwinding; t, time) using GraphPad Prism.

**CryoEM analysis.** Complexes were prepared freshly in 20 mM Tris-HCl, pH 8.0, 150 mM NaCl, 1 mM DTT and concentrated to 5 mg/ml using ultrafiltration. Immediately before grid preparation, the samples were supplemented with 0.15% (w/v) n-octylglucoside to overcome preferred particle orientation. 3.8 μl of the final mixtures were applied to plasma-treated R1.2/1.3 holey carbon grids (Quantifoil Micro Tools GmbH). Grids were plunged into liquid ethane after blotting, using a Vitrobot Mark IV device (FEI) at 10 °C/100% humidity. Image acquisition was conducted on a FEI Titan Krios G3i (300 kV) with a Falcon 3EC camera, operated in counting mode using EPU software (Thermo Fisher Scientific). The BRR2[HR]-

PRPF8[Jab1]-C9ORF78 dataset was acquired with a nominal magnification of 120,000x, resulting in a pixel size of 0.657 Å/px. The BRR2[HR]-FBP21[200–376] dataset was acquired with a nominal magnification of 92,000x, resulting in a pixel size of 0.832 Å/px. A total electron dose of 40 e/Å² was accumulated over an exposure time of 31 s and 40 s, respectively.

All image analysis steps were done with cryoSPARC. Movie alignment was done with patch motion correction, CTF estimation was conducted by Patch CTF. For the BRR2[HR]-FBP21[200–376] dataset, class averages of manually selected particles were used to generate an initial template for reference-based particle picking from 1,877 micrographs. Particle images were extracted with a box size of 336 px and binned to 84 px for initial analysis. Ab initio reconstruction using a small subset of particles was conducted to generate an initial 3D reference for 3D heterogeneous refinement. 271,493 selected particle images were re-extracted with a box of 168 px (1.664 Å/px) and subjected to non-uniform refinement followed by heterogeneous refinement. Finally, 165,220 particle images were re-extracted using local motion correction at full spatial resolution (box size 336 px). After per-particle CTF correction, non-uniform refinement was applied to generate the final reconstruction at a resolution of 3.3 Å.

The BRR2[HR]-PRPF8[Jab1]-C9ORF78 dataset was refined similarly with only minor differences. References generated from the BRR2[HR]-FBP21[200–376] dataset were used for picking and refinement. To improve the density for C9ORF78, 3D variability analysis was applied using a mask generously enclosing the N-terminal region of C9ORF78. From 542,565 particle images, a total of 370,493 particle images were selected for the final non-uniform refinement, yielding a reconstruction at 2.76 Å resolution.

**Model building and refinement.** The final cryoEM map for the BRR2[HR]-PRPF8[Jab1]-C9ORF78 complex was used for placement of the BRR2[HR] model in complex with PRPF8[Jab1] [25] employing PHENIX Dock in Map[66]. The additional density observed after model placement was of sufficient quality to manually and unequivocally build the N-terminal residues 5–58 of C9ORF78. The region spanning residues 12–23 exhibited poorer cryoEM density and was modelled with less reliability. An additional refined map focusing on C9ORF78 displayed an improved density for this area providing more confidence in modeling. The structure of BRR2[HR] was placed into the final cryoEM map of the BRR2[HR]-FBP21[200–376] complex. Parts of FBP21[200–376] could be unequivocally built into additional density that was not covered by the BRR2[HR] model. Models were completed and manually adjusted residue-by-residue, supported by real space refinement in Coot. The manually built models were refined against the cryoEM maps using the real space refinement protocol in PHENIX. The structures were evaluated with Molprobity[67]. Structure figures were prepared using PyMOL (Version 1.8 Schrödinger, LLC).

**Culturing HEK293T cells and transient transfection.** HEK293T cells were grown in DMEM supplemented with 10% (v/v) fetal bovine serum and 1% (w/v) penicillin/streptomycin (Invitrogen). Transient transfection was performed using Rotifect (Carl Roth) according to the manufacturer's instructions. Briefly, $4.5 \times 10^5$ cells were seeded on 6-well plates 24 h prior to transfection. For each well to be transfected, 2 μg plasmid and 5 μl Rotifect were mixed with 250 μl Optimum, and incubated for 5 min at room temperature. Reactions were mixed, incubated for 20 more minutes at room temperature and then added to the cells. Cells were harvested 48 h after transfection.

**Nuclear extract preparation.** HEK293T or Flp-In™ T-REx™ 293 cells grown in T75 flasks were harvested in cold PBS and pelleted by centrifugation ($1000 \times g$, 1 min). Cell pellets were resuspended in 600 μl cold CTX buffer (10 mM HEPES-NaOH, pH 7.9, 1.5 mM MgCl₂, 10 mM KCl), supplemented with proteinase inhibitors, and incubated for 5 min on ice. A total of 600 μl CTX supplemented with 0.2 (v/v)% NP-40 were added, and the reactions, after gentile mixing, were incubated for another 5 min on ice. Nuclei were pelleted by centrifugation at $4000 \times g$ in a table-top centrifuge for 3 min, and the supernatant (cytosolic fraction) was discarded. Nuclei were resuspended in 240 μl NX buffer (20 mM HEPES-NaOH, pH 7.9, 1.5 mM MgCl₂, 420 mM KCl, 0.2 mM EDTA, 25% (v/v) glycerol), supplemented with proteinase inhibitors, and subsequently three times frozen (−80 °C) and thawed (37 °C), followed by 1 minute of vortexing. After a final centrifugation step at maximum speed for 20 min at 4 °C, the supernatant (nuclear extract) was stored at −20 °C until further use. Protein concentrations were determined via Bradford assay.

**Flag-IP combined with Western blot.** For Flag-IP combined with Western blot, 100 μg nuclear extract were incubated with 400 μl RIPA lysis buffer, including 100 mM NaCl, 2% (w/v) BSA and proteinase inhibitors, for 1 h at 4 °C on a rotating wheel. Subsequently, 15 μl Flag M2 affinity gel (Sigma Aldrich), equilibrated in RIPA buffer, were added to the reactions and incubated overnight at 4 °C on a rotating wheel. The resin was washed four times with RIPA buffer (without BSA) with centrifugation at $4000 \times g$ for 1 min, then resuspended in 40 μl SDS-loading dye (2-fold concentrated). Samples were boiled at 96 °C for 5 min, centrifuged at maximum speed for 2 min, and the supernatant was then analyzed by SDS-PAGE and Western blot using monoclonal anti-Flag M2 antibody (Sigma Aldrich),

diluted 1:2000, and rabbit serum containing a polyclonal anti-human BRR2 antibody, diluted 1:500.

**Generating stable cell lines.** Stable Flp-In™ T-REx 293 cells for expression of C9ORF78 and C9ORF78[R41A] with C-terminal 3xFlag-His$_6$-Biotin-His$_6$ (3xFlag-HBH) tags were generated as described before[52]. Transfection of the cell lines were done using Lipofectamine 2000 (Thermo Fisher Scientific). After hygromycin selection, expression of the tagged proteins was confirmed by Western blot using monoclonal anti-Flag M2 antibody.

**Flag-IP followed by mass spectrometry.** For mass spectrometric analysis of C9ORF78 interactors, Flp-In™ T-REx™ 293 cells stably expressing C-terminally 3xFlag-HBH-tagged C9ORF78 or C9ORF78[R41A] were grown in T75 flasks, and nuclear extracts were prepared as described above. As control, unmodified Flp-In™ T-REx™ 293 cells were used. A total of 500 µg nuclear extract were incubated with 800 µl IP buffer (10 mM HEPES-NaOH, pH 7.3, 150 mM NaCl,10 mM MgCl$_2$, 10 mM KCl, 0.5 mM EGTA), supplemented with 3 U/ml benzonase and proteinase inhibitors, for 1 h on a rotating wheel. For each IP, 50 µl Flag M2 affinity gel (Sigma Aldrich), equilibrated in IP buffer, were added to the reactions and incubated overnight at 4 °C on a rotating wheel. The resin was washed four times with IP buffer (without supplements) by centrifugation at 4000 × g for 1 min. Bound proteins were eluted by incubation with 50 µl 3xFlag Peptide (Sigma Aldrich) at a concentration of 0.5 µg/µl in TBS (50 mM Tris-HCl, pH 7.5, 150 mM NaCl) for 30 min on ice. The supernatant was run on a SDS-PAGE gel until entrance into the separating gel. Bands were excised and digested with trypsin using a standard protocol[68]. After digestion, peptides were extracted and dried for LC-MS analysis.

Peptides were reconstituted in 10 µl of 0.05% (w/v) TFA, 2% (v/v) acetonitrile, and 7 µl were applied to an Ultimate 3000 reversed-phase capillary nano liquid chromatography system connected to a Q Exactive HF mass spectrometer (Thermo Fisher Scientific). Samples were injected and concentrated on a PepMap100 C18 trap column (3 µm, 100 Å, 75 µm i.d. × 2 cm; Thermo Fisher Scientific) equilibrated with 0.05% (w/v) TFA in water. After switching the trap column inline, LC separations were performed on an Acclaim PepMap100 C18 capillary column (2 µm, 100 Å, 75 µm i.d. × 25 cm; Thermo Fisher Scientific) at an eluent flow rate of 300 nl/min. Mobile phase A contained 0.1% (v/v) formic acid in water, and mobile phase B contained 0.1% (v/v) formic acid in 80% (v/v) acetonitrile. The column was pre-equilibrated with 5% mobile phase B followed by an increase to 44% mobile phase B over 100 min. Mass spectra were acquired in a data-dependent mode, utilizing a single MS survey scan (m/z 350–1650) with a resolution of 60,000 in the Orbitrap, and MS/MS scans of the 15 most intense precursor ions with a resolution of 15,000. The dynamic exclusion time was set to 20 s, and the automatic gain control was set to $3 \times 10^6$ and $1 \times 10^5$ for MS and MS/MS scans, respectively.

MS and MS/MS raw data were analyzed using the MaxQuant software package with implemented Andromeda peptide search engine[69]. Data were searched against the human reference proteome downloaded from Uniprot (77,027 proteins, taxonomy 9606, last modified January 29, 2021) using label-free quantification, and the match between runs option was enabled. Filtering and statistical analysis was carried out using Perseus software[70]. Only proteins which were identified with LFQ intensity values in all three replicates (within at least one of the 3 experimental groups) were used for downstream analysis. Missing values were replaced from normal distribution (imputation), using the default settings (width 0.3, down shift 1.8). Mean log2-fold differences between Flag-C9ORF78 IP (wt or R41A) against control IP were calculated in Perseus, using Student's t-tests with permutation-based FDR of 0.05.

**siRNA KD and rescue.** For siRNA KD of C9ORF78, $2.25 \times 10^5$ HEK293T cells per well were seeded on 6-well-plates in 1.5 ml/well DMEM supplemented with 10% (v/v) fetal bovine serum (without antibiotics) one day prior to transfection. For each KD three technical replicates were prepared. Transfection was carried out as described above, using 10 µl of 20 µM siRNA instead of plasmid (1:1 mix of siRNA1 5′-GCA ACU GAU GAC UAU CAU UTT-3′ and siRNA2 5′-CCA GAG AGG UAC AGA ACU UTT-3′; Eurofins Genomics). Cells were harvested in cold PBS 72 h post transfection, pelleted and resuspended in 1 ml RNA Tri-flüssig (Bio&Sell) for RNA extraction. RNA was extracted following the manufacturer's instructions. RNA pellets were dissolved in 88 µl water and subsequently digested with DNase I for 20 min at 37 °C, followed by RNA extraction with ROTI-Aqua-P/C/I (Carl Roth). RNA concentrations were quantified on a Nanodrop spectrophotometer and adjusted to 0.5 µg/µl. For rescue experiments, cells were transfected with either empty pcDNA3.1 (+) or siRNA-resistant C9ORF78 (wt or R41A) genes in pcDNA3.1(+) vectors (ThermoFisher) 2 h prior to transfection with siRNAs, and harvested after 72 h.

**RT-PCR, quantitative PCR and radioactive PCR.** For reverse transcription-PCR (RT-PCR), 1 µg RNA was used with 1 ng gene-specific reverse primer (combining up to 3 gene specific primers in one RT reaction; Supplementary Table 2). The reaction was carried out using M-MuLV reverse transcriptase (Enzymatics) according to the manufacturer's protocol.

Quantitative PCR (qPCR) was performed in 96 well format using the PowerUp SYBR Green Master Mix (Thermo Fisher) on a Stratagene Mx3000P instruments. qPCR reactions were performed in duplicates. Mean values were used to normalize expression to mRNA of GAPDH(ΔCT). Values shown are means ± standard deviation, p-values were calculated using Student's unpaired t-test (Microsoft Office, Excel). Significance is indicated by asterisks ($*p < 0.05$; $**p < 0.01$; $***p < 0.001$).

Low-cycle PCR with a [$^{32}$P]-labeled forward primer (Supplementary Table 2) was performed using Q5 High Fidelity DNA Polymerase (NEB). Products were separated by denaturing 5% PAGE and quantified using a Phosphoimager and ImageQuantTL software. Quantifications are given as mean values of several technical and biological replicates, error bars represent standard deviation, p-values were calculated using Student's unpaired t-test (Microsoft Office, Excel). Significance is indicated by asterisks ($*p < 0.05$; $**p < 0.01$; $***p < 0.001$; $****p < 0.0001$).

**RNA sequencing and data analysis of siRNA KD.** RNA sequencing was performed in technical triplicates using DNase I-digested RNA samples for library preparation. Libraries were prepared using the mRNA enrichment method at BGI Genomics and sequenced using DNBSeq PE150 sequencing. This yielded ~35 million paired-end 150 nt reads for C9ORF78 and control samples. Reads were aligned to the human hg38 genome, using STAR, yielding ~ 95% uniquely aligned reads. Alternative splicing changes were calculated using rMATS. To obtain only high confidence targets, only targets with a ΔPSI > 0.15 and a p-value < 0.01 were considered alternatively spliced. Additionally, to filter out splicing events in weakly expressed genes or gene regions with low expression, events with <100 combined junction reads in all samples were excluded. RNA sequencing after KD/rescue were analyzed identically. To find consistently changed exons in the two independent si control and si C9ORF78 triplicate samples, we changed the filters to ΔPSI > 0.1 and p-value < 0.05.

To analyze the splice site strength of altered exons, the respective coordinates were extracted from the rMATS output table in bed format (-20 to +3 for 3′-ss; -3 to +6 for 5′-ss). Bedtools getfasta was then used to extract splice site sequences. Sequences were input to MaxEntScan[71], using the Maximum Entropy Model to score. As a control, all rMATS-quantified exons were treated identically.

**FLASH.** FLASH was carried out essentially as described before[52]. Briefly, Flp-In™ T-REx™ 293 cells expressing either 3xFlag-BIO tagged C9ORF78 or 3xFlag-BIO tagged GFP were induced with 0.1 µg/mL doxycycline for ~16 h, and UV-crosslinked with 0.2 mJ/cm$^2$ UV-C light. Target proteins were purified using MyONE C1 streptavidin beads (ThermoFisher Scienific). After a partial RNase digestion using RNase I and end-repair with T4 PNK, uniquely barcoded s-oligos were ligated to each sample (two biological replicates). After several stringent washes (up to 1% (w/v) SDS), biological replicates were merged, bead-bound RNA was released with proteinase K and purified with Oligo Clean and concentrator columns (Zymo Research). RNA was then reverse-transcribed and treated with RNase H to phosphorylate 5′-ends of cDNA. The cDNA was circularized with CircLigaseII (Lucigen), amplified with Q5 polymerase and sequenced on an Illumina platform (100 bp, PE sequencing).

The resulting fastq files were first merged with bbmerge, replicates were then split using flexbar and mapped to hg38 assembly of the human genome using bowtie2 and bbmap, after which umi-tools was used to remove PCR duplicates. Enrichment plots were generated using the snakePipes noncoding-RNA-seq workflow, which uses TEtools to calculate specific enrichment of non-coding RNAs against a background dataset (GFP in this case). Cumulative coverage plots were generated with computeMatrix scale-regions and plotHeatmap from the deepTools2 package, using alternatively spliced exons and 3′-ss detected with rMATS.

**Reporting summary.** Further information on research design is available in the Nature Research Reporting Summary linked to this article.

## Data availability

The data that support this study are available from the corresponding author upon reasonable request. CryoEM maps have been deposited in the Electron Microscopy Data Bank (https://www.ebi.ac.uk/pdbe/emdb) under accession codes EMD-13046 for BRR2[HR]-PRPF8[Jab1]-C9ORF78, and EMD-13045 for BRR2[HR]-FBP21[200-376]. Structure coordinates have been deposited in the RCSB Protein Data Bank (https://www.rcsb.org) with accession codes 7OS2 for BRR2[HR]-PRPF8[Jab1]-C9ORF78 (Ref. [72]) and 7OS1 for BRR2[HR]-FBP21[200-376] (Ref. [73]). RNA sequencing data from siRNA KD and rescue experiments have been deposited in Gene Expression Omnibus (GEO) under accession codes GSE176517 and GSE189362. FLASH data have been deposited in GEO under accession code GSE176464. The mass spectrometry proteomics data have been deposited in the ProteomeXchange Consortium via the PRIDE[74] partner repository (https://www.ebi.ac.uk/pride/) under dataset identifier PXD031482. The reference proteome for the analysis of the mass spectrometry proteomics data was obtained from Uniprot (https://www.uniprot.org/proteomes/UP000005640). All other data are contained in the manuscript or the Supplementary Information. Source data are provided with this paper.

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

## Acknowledgements

Large-scale yeast two-hybrid analyses were conducted in collaboration with the group of Ulrich Stelzl, University of Graz, Austria, in collaboration with Camilla von Alvensleben, Laboratory of Structural Biochemistry, Freie Universität Berlin. We acknowledge the assistance of the core facility BioSupraMol supported by the Deutsche For-schungsgemeinschaft in mass spectrometric and electron microscopic analyses. This work was supported by grants from the Deutsche Forschungsgemeinschaft (TRR186-A15 to F.H. and M.C.W.; INST 130/1064-1 FUGG to Freie Universität Berlin) and the Berlin University Alliance (501_BIS-CryoFac to M.C.W.). Work in the group of T.A. was supported by the Max Planck Society.

## Author contributions

A.B. cloned genes, purified proteins, generated stable cell lines with help of T.A. and I.A.I. and performed all experiments, except for sample preparation for mass spectrometry (B.K.), grid freezing for cryoEM (T.H.) and FLASH sample preparation of UV-crosslinked cells (I.A.I.). M.P. analyzed RNA-seq data derived from the siRNA KD experiments and contributed to experimental design. B.K. acquired and processed pro-teomics data. I.A.I. analyzed RNA-seq data derived from the FLASH experiments. T.H. acquired, processed, and refined cryoEM data. M.C.W. built atomic models and G.W. refined structures. All authors contributed to the analysis of the data and the inter-pretation of the results. A.B. wrote the manuscript with contributions from the other authors. C.F., T.A., F.H., and M.C.W. supervised work in their respective groups. M.C.W. conceived and coordinated the project.

## Funding

## Competing interests

The authors declare no competing interests.
