## [Peer Review File · Nature Communications]

A multi-factor trafficking site on spliceosome remodeling enzyme, BRR2, recruits C9ORF78 to regulate alternative splicingReviewers' Comments:

Reviewer #1:

Remarks to the Author:

The manuscript by Bergfort et al., employs methods of structural and molecular biology to study the role of poorly characterised C9ORF78 protein in splicing. C9ORF78 has been previously reported as a constituent part of spliceosome whose function remained largely unknown. The manuscript reports a cryoEM structure of C9ORF78 bound to its interaction partner BRR2. The second cryoEM structure, reported in the manuscript is that of BRR2 bound to an inhibitor FBP21. These structures together with pulldown experiments allowed authors to conclude that C9ORF78 and FBP21 binding to BRR2 is mutually exclusive. Armed with that knowledge, the authors then went on to investigate the effect of C9ORF78 knockdown on mRNA splicing. siRNA knockdown of C9ORF78 had an effect on 3' splice site selection which induced exon skipping events, or altered exon length if the two 3' splice sites were in close proximity.

This structural knowledge allowed the authors to pinpoint C9ORF78 residues, critical for interaction with BRR2. The fact that these residues are conserved over hundreds millions of years of evolution and the corresponding point mutant C9ORF78-R41A fails to bind to BRR2 is a highlight of the paper. The place of C9ORF78 in splicing is further cemented by UV crosslinking experiments that show a link between C9ORF78 and U5 snRNA. These experiments however did not reveal significantly enriched mRNAs, suggesting that C9ORF78 does not affect splicing via direct binding to mRNAs, and rather exerts its influence on the 3' splice site selection via its spliceosome interaction partners.

This body of work is novel and represents a significant interest to the field of splicing and broader readership. The structures presented in the paper are of particular value to the field as C9ORF78 has not been observed in the published spliceosomal structures. However I have 1 major and several minor comments that I hope would help the authors improve their manuscript further.

Major point

The manuscript presents detailed structural description of how C9ORF78 interacts with BRR2 and PRPF8 proteins. However, experimental support for functional significance of these interactions is perhaps the weakest part of otherwise solid story the authors portray. Indeed, this support becomes vital, as the authors report that more than a couple of thousand proteins are enriched over background in C9ORF78 IP. This would suggest that in addition to a direct interaction with BRR2, C9ORF78 could influence splicing indirectly via its non-spliceosomal interaction partners. The functional validation experiment, shown in figure 5b has produced mixed results. C9ORF78-WT and C9ORF78-R41A rescue si RNA mediated C9ORF78 knockdown to identical extents (figure 5b SMARCA4 and C10RF131 panels). For the experiment in panel PTBP2, the authors compared C9ORF78-WT knockdown rescue to that of C9ORF78-R41A and reported a difference between the two. Firstly, that comparison may not be the right one as C9ORF78-WT "rescues" exon skipping to the levels better than in si-Control. Why does it do that is an unsolved question, perhaps C9ORF78 is limiting and its overexpression produces better splicing efficiency in si-C9ORF78 cells than in unperturbed cells that do not express si resistant C9ORF78. Therefore, as far as splicing efficiency is concerned, a comparison between PTBP2 exon 10 retention in si-Control against a C9ORF78 knockdown rescued by C9ORF78-R41A is the one to make in this case. It would appear as if this comparison would not produce a statistically significant difference between the two. If that is correct, then we would have to conclude that splicing of PTBP2 exon 10 is indistinguishable in si-Control vs si-C9ORF78 rescued with C9ORF78-R41A, hence C9ORF78 interaction with BRR2 is dispensable for splicing of that particular exon.

Even if the authors disagree with this logic, I would draw their attention to absolute fold change values between si-Control and C9ORF78-R41A they report in panel PTBP2 of figure 5. The reported difference in fold change is miniscule. How would that translate into protein level and what effect would it have on the function of PTBP2? From figure 5 I concluded that the effect would negligible at best.

Therefore I would strongly urge the authors to find a clear cut example for functional significance of C9ORF78-BRR2 interaction. A good starting point would be to expand the experimental approach and analysis reported in figure 4 onto figure 5 and perform the necessary RNA-seq experiments with the samples of Figure 5.

Minor points:

1. Figure 2a C9ORF78 labelling – there are 2 alpha 1 and no alpha 3 present.
2. “We exchanged C9ORF78 F8 and R41 individually for alanine residues, and tested BRR2 binding of the C9ORF78 variants via analytical SEC. While C9ORF78F8A showed reduced binding to BRR2 in analytical SEC, BRR2 binding by C9ORF78R41A in vitro was completely abolished”

C9ORF78-F8A is not present in Figure 3b, perhaps the authors have forgotten to add the panel with WB analysis of a SEC experiment for the C9ORF78F8A mutant.

3. Figure 4c. The first 2 panel appear to be only depicting statistically significant splicing events, while the NAGNAG panel appears to show all events. Perhaps the authors could consider showing only statistically significant NAGNAG events similarly to the previous 2 panels of the figure.

4. Figure 5a. Labelling of y axis. It would be beneficial to explain in the text or figure legend what PSI is and how it is calculated.

5. Figure 5b. It would be beneficial to explain in the text or figure legend how the fold change was calculated.

6. Figure 6a. The authors could look at their FLASH data for sites of crosslink between C9ORF78 and U5 snRNA. With FLASH I would expect that an RNA-protein crosslink site would manifest itself as a tight cluster of deletions or point mutations in cDNA sequences. If it is possible to elucidate the exact crosslink site, it could significantly contribute to the structural data and perhaps could allow further speculations on the location of the part of C9ORF78 that is not engaged in interaction with BRR2.

7. Comments on the C9ORF78 interactome.

The authors show that spliceosomal proteins are enriched in in both C9ORF78-WT and C9ORF78-R41A. It would be good to see a GO term analysis, considering the manuscript reports that C9ORF78 IP results in enrichment over control for 2411 proteins. This is perhaps 1/5th of the proteome, expressed in HEK cells. In fact, the very limited GO analysis that I performed on the manuscripts data does reveal a strong enrichment for GO CC “spliceosome”, which helps interpretation of the data, as it shows that the IP has been specific.

To that end, I would suggest that in addition to figure 6b the chapter “C9ORF78 interacts with additional spliceosomal proteins” could benefit from a figure that would depict the proteins, belonging to two spliceosomal complexes B and C. Those proteins should be colour coded to depict 2 key parameters, discussed in the chapter: whether a given protein was detected as enriched in the IP and if so, whether its enrichment is higher or lower in C9ORF78-WT compared to C9ORF78-R41A. I am of the opinion that this would be a great visual help for the chapter. Perhaps the authors could consider including a figure like that in the manuscript.

Reviewer #2:

Remarks to the Author:

The manuscript by Bergfort et al reports an unstructured protein C9ORF78 tightly interacts with the key spliceosomal RNA helicase BRR2 through a series of evidence, including yeast two-hybrid screen, in vitro protein-protein interaction, affinity purification and mass spectrometry, and cryoEM structures. They find that C9ORF78 and another spliceosomal protein FBP21 interact with the C-terminal cassette of BRR2 in a mutually exclusive manner using both structural information and biochemical competition assay. RNAi of C9ORF78 leads to alternative splicing changes including a substantial usage of alternative 3'SSs. This manuscript provides insightful information in understanding the function of a flexibly or dynamically bound spliceosomal protein during the process of spliceosome assembly and catalysis. In general, their findings are convincing and interesting, and the manuscript is well written.

Below are several questions and concerns:

1. Typo: In Figures 1-3, several places of C9ORF78 are shown in "C9ORF8"; In Figures 2a and 3a, there are two α 1s labelled for the structure of C9ORF78, of which one should be " α 3"; In Figure 7, consistent with its legends, the forest green component marked as "PRP22" should be "PRPF22".
2. C9ORF78 was observed in the C complex and the binding of C9ORF78 with BRR2 was also described in *S. pombe*. In Figures 1a & 1b and later, overexpression of GST- or Flag-tagged C9ORF78 are used for SEC and IP experiments, indicating that the interaction between C9ORF78 and BRR2 might be not strong in HEK293 cells. Could this be done using an endogenous normally expressed C9ORF78, either through an C9ORF78-specific antibody or by a CRSIPR-Cas9 knock-in tag system?
3. In Figure 4a, this should be RT-qPCR, not qPCR, detection of mRNAs, better to present this with an additional agarose gel.
4. In Figure 4b, SE (skipped exon) is the dominant feature of AS events when the C9ORF78 is KD. I am curious, what is the feature of those exons? For those increased inclusion of exons (up-regulated), do they have weaker 3'SSs; vice versa, for those decreased inclusion of exons (down-regulated), do they have stronger 3'SSs? Therefore, I would like to suggest strength analyses (scores) of both the 3'SSs and 5'SSs of those SE events.
5. In Figure 4c, the negative value of deltaPSI are not presented, this is confusing of which sample vs which sample. In addition, RMATS should be rMATS.
6. The primes in "3' or 5'-splice site" are incorrect, should be 3' or 5'.

Reviewer #3:

Remarks to the Author:

In this manuscript the authors present a rather comprehensive analysis of the interactions of the human splicing factor C9ORF78 with the spliceosome and particularly with the helicase Brr2, revealing a previously unappreciated role for this factor during catalysis in modulating 3'SS selection. The manuscript suggests a molecular mechanism that can explain how Brr2 may act during the catalytic stage.

I generally find the authors' data compelling, of timely interest to the field, and potentially more broadly, while the proposed mechanistic models are mostly supported by the presented experiments. However, there are a few points that the authors should try to address prior to publication.

While the cryo-EM analysis appears sound and expertly performed, the authors do not provide any clear figure showing their fit of the C9ORF78 model into their determined EM map. Given the claimed high resolution, it is critical for the authors to show this data as a figure in the paper, especially for the critical parts where C9ORF78 interacts with Brr2. The mutational data in Fig. 3 does support the proposed modelling, but still it is impossible to fully judge the quality of the map and the model fit

without this data, especially as the presented local resolution in Sup.Fig. 2e shows significant variation in local resolution along the proposed C9ORF78 path. Certainly the map presented in Fig. 1c is contoured at an RMSD that makes it hard to judge the presence of high resolution features.

Although the in vitro assays show a modest effect of C9ORF78 on Brr2 helicase activity for U4/U6, I think it is premature to argue that C9ORF78 does not act through modulation of Brr2 helicase activity, as the authors do on p.8. The assays used are not the native situation in the spliceosome and C9ORF78 binds at the U2/U6 stage of splicing, while Brr2 has been implicated in spliceosome disassembly. Thus, one could easily imagine that C9ORF78 may act at the disassembly stage and that its role in 3'SS selection could be coupled to a role in modulating Brr2 activity during disassembly. I am aware that this role for Brr2 is a matter of contention in the field but I think the authors should be more cautious with their statements here, as their data cannot exclude a role for C9ORF78 in modulating Brr2 during this later stage.

The observations regarding competition between FBP21 and C9ORF78 for Brr2 binding are strongly supported by the data in Fig. 2d. Nonetheless, it is unclear why the authors chose to use Brr2HR complexes rather than complexes that also contained the Jab1 domain of Prp8, given that the Jab1 domain remains bound to Brr2 from the B complex onwards and that the FBP21 interaction is observed in the B complex structure in the context of Brr2 being bound to Jab1. I think the authors could and likely should strengthen their argument here by performing the competition experiment with Brr2HR/Jab1/C9ORF78 complexes and even better do it in the proper physiological progression by asking whether C9ORF78 can compete off FBP21, which is what they propose actually happens during the splicing pathway.

Finally, while I appreciate the authors' model for C9ORF78 function during the C* stage as a compelling main mechanism of action, I think the authors need to be much more careful with their discussion of various proteins they claim are present at the C complexes stage, such as DDX23 or Prp6. These have generally only been detected as such in C complexes prepared by biochemical stalls that are not entirely clean and could have contaminating earlier and later complexes. A good example are studies that use a 3'SS mutation to capture C complexes and which are now known from cryo-EM studies to actually reach the C* and even P complex stage and then revert back to the C complex stage. Simply citing proteomic analyses of such preparations as evidence of stage-specific association is sufficient evidence for such claims. None of the single particle EM studies of C or C* complexes have identified any subpopulations containing DDX23 for example, yet the authors routinely seem to suggest this protein could bind at this stage. The claim that their IP of DDX23 occurs at the C complex stage is particularly problematic in this sense, especially as one could imagine much more easily how complexes that have transitioned to the B complex stage may not have fully lost DDX23 until the Bact stage, making the exchange of FBP21 for C9ORF78 an alternative point at which a transient interaction between C9ORF78 and DDX23 could have occurred. Similarly, the claim that Prp22 binds at the C complex stage should be revised, as what people have reported as binding at that stage most likely results from 3'SS mutant complexes that have reached the C* stage and then reverted back to the C complex stage without Prp22 dissociation, as was shown in a recent study on equilibrium of spliceosome conformations during catalysis in yeast. I urge the authors to be much more rigorous with their terminology when describing potential binding to various complexes.

Related to this matter, I find that the authors overlook too quickly a potential role for C9ORF78 in regulating splice site use also during the B to Bact transition. They observe significant numbers of exon skipping and mutually exclusive exon use changes in their KD experiments. It is much harder to imagine how such events could be regulated at the C or C* stage, but much easier to imagine how transfer of the 5'SS and docking of the BP helix at the active site, which occur during the B to Bact transition, or are influenced by the relative stability of these complexes, could impact these types of alternative splicing events. Indeed, the crosslinks to U5 snRNA are consistent with such a role and much more likely to reflect interactions at the Bact stage than at the C complex stage, when U5 is buried very deeply into the active site, making an interaction with a flexible part of C9ORF78 harder to

imagine. Brr2-dependent association with Cwc27 also supports this idea, as in the Bact structure the flexible C9ORF78 residues proposed to interact with Prp22 in C*, could easily be imagined to interact with Cwc27 in Bact. Since C9ORF78 likely regulates the Bact transition, as the authors argue with strong experimental support, they should at least discuss the possibility of an earlier role in regulating alternative splicing at this stage through some of the other factors they observe in their IP studies.

Response to Reviewer Comments

Reviewer comments are repeated in bold italics, responses are in regular font, changed text passages are highlighted in yellow.

Reviewer #1:

The manuscript by Bergfort et al., employs methods of structural and molecular biology to study the role of poorly characterised C9ORF78 protein in splicing. C9ORF78 has been previously reported as a constituent part of spliceosome whose function remained largely unknown. The manuscript reports a cryoEM structure of C9ORF78 bound to its interaction partner BRR2. The second cryoEM structure, reported in the manuscript is that of BRR2 bound to an inhibitor FBP21. These structures together with pulldown experiments allowed authors to concluded that C9ORF78 and FBP21 binding to BRR2 is mutually exclusive. Armed with that knowledge, the authors then went on to investigate the effect of C9ORF78 knockdown on mRNA splicing. siRNA knockdown of C9ORF78 had an effect on 3' splice site selection which induced exon skipping events, or altered exon length if the two 3' splice sites were in close proximity.

This structural knowledge allowed the authors to pinpoint C9ORF78 residues, critical for interaction with BRR2. The fact that these residues are conserved over hundreds millions of years of evolution and the corresponding point mutant C9ORF78-R41A fails to bind to BRR2 is a highlight of the paper. The place of C9ORF78 in splicing is further cemented by UV crosslinking experiments that show a link between C9ORF78 and U5 snRNA. These experiments however did not reveal significantly enriched mRNAs, suggesting that C9ORF78 does not affect splicing via direct binding to mRNAs, and rather exerts it's influence on the 3' splice site selection via its spliceosome interaction partners.

This body of work is novel and represents a significant interest to the field of splicing and broader readership. The structures presented in the paper are of particular value to the field as C9ORF78 has not been observed in the published spliceosomal structures. However I have 1 major and several minor comments that I hope would help the authors improve their manuscript further.

We thank the reviewer for this very positive overall assessment, specifically for considering our work to be of significant interest to the field and beyond, and our structures to be of particular value.

Major point

The manuscript presents detailed structural description of how C9ORF78 interacts with BRR2 and PRPF8 proteins. However, experimental support for functional significance of these interactions is perhaps the weakest part of otherwise solid story the authors portray. Indeed, this support becomes vital, as the authors report that more than a couple of thousand proteins are enriched over background in C9ORF78 IP. This would suggest that in addition to a direct interaction with BRR2, C9ORF78 could influence splicing indirectly via it's non-spliceosomal interaction partners.

The functional validation experiment, shown in figure 5b has produced mixed results. C9ORF78-WT and C9ORF78-R41A rescue si RNA mediated C9ORF78 knockdown to identical extents (figure 5b SMARCA4 and C1ORF131 panels). For the experiment in panel PTBP2, the authors compared C9ORF78-WT knockdown rescue to that of C9ORF78-R41A and reported a difference between the two.

Firstly, that comparison may not be the right one as C9ORF78-WT "rescues" exon skipping to the levels better than in si-Control. Why does it do that is an unsolved question, perhaps C9ORF78 is limiting and it's overexpression produces better splicing efficiency in si-C9ORF78 cells than in unperturbed cells that do not express si resistant

C9ORF78. Therefore, as far as splicing efficiency is concerned, a comparison between PTBP2 exon 10 retention in si-Control against a C9ORF78 knockdown rescued by C9ORF78-R41A is the one to make in this case. It would appear as if this comparison would not produce a statistically significant difference between the two. If that is correct, then we would have to conclude that splicing of PTBP2 exon 10 is indistinguishable in si-Control vs si-C9ORF78 rescued with C9ORF78-R41A, hence C9ORF78 interaction with BRR2 is dispensable for splicing of that particular exon. Even if the authors disagree with this logic, I would draw their attention to absolute fold change values between si-Control and C9ORF78-R41A they report in panel PTBP2 of figure 5. The reported difference in fold change is miniscule. How would that translate into protein level and what effect would it have on the function of PTBP2? From figure 5 I concluded that the effect would negligible at best. Therefore I would strongly urge the authors to find a clear cut example for functional significance of C9ORF78-BRR2 interaction. A good starting point would be to expand the experimental approach and analysis reported in figure 4 onto figure 5 and perform the necessary RNA-seq experiments with the samples of Figure 5.

While we generally agree with the reviewer's assessment, we would like to point out that due to the single point mutation we introduced into the C9ORF78^{R41A} variant (which abrogates stable interaction with BRR2 *in vitro* in the absence of other factors but may be partly "overridden" by additional interactions of C9ORF78 with other components of spliceosomes, as suggested by our IP experiments) and unavoidable overexpression of the siRNA-resistant C9ORF78^{wt} or C9ORF78^{R41A} variants for rescue experiments, full differential rescue effects by the C9ORF78^{wt} or C9ORF78^{R41A} variants are not to be expected.

We thank the reviewer for the suggestion of conducting KD/rescue experiments in combination with RNA-seq, and took it up. In the course of these additional experiments, we had to conduct a second, independent KD experiment, which allowed us to further validate our observations from the first KD experiment.

Importantly, the second, independent KD experiment confirmed the findings of the first KD experiment: (i) increased skipping of upstream 3'-ss in alternative 3'-ss pairs with a strong overlap in the affected 3'-ss pairs; (ii) a significant number of affected exon skipping events, albeit with a lower overlap between the two KD events. Consistent with our previous experiments, over-expression of siRNA-resistant variants of both C9ORF78^{wt} and C9ORF78^{R41A} globally reverted C9ORF78 KD-dependent changes in alternative 3'-ss usage, unequivocally confirming that the observed effects are specific to C9ORF78, but failing to reveal a direct role of the observed BRR2-C9ORF78 interaction in these events. However, 49 exon skipping events, which were reproducibly altered upon C9ORF78 KD in the two independent KD experiments, were significantly rescued only by over-expression of C9ORF78^{wt}, but not by over-expression of C9ORF78^{R41A}. PTBP2 exon 10 skipping was among the exon skipping events reproducibly affected in both independent KD experiments and was only significantly rescued by over-expression of C9ORF78^{wt}. Furthermore, in the new, global RNA-seq based assessment of KD/rescue, C9ORF78^{wt} over-expression did not restore PTBP2 exon 10 skipping levels "beyond" the si control situation. We now also statistically compare si control and rescue by C9ORF78^{R41A}, as suggested (new Fig. 5g).

Quantifying the effect of C9ORF78 KD on PTBP2 exon 10 skipping based on the new KD/rescue analysis, revealed that C9ORF78 KD led to a decrease in the level of PTBP2 exon 10 inclusion from ~ 62 % spliced-in for si control to ~50 % spliced-in for si C9ORF78 (new Fig. 5g). While we agree that this effect is moderate, we note that the PTBP2 isoform obtained upon exon 10 skipping is an NMD target, which might diminish the observable effect.

Taken together, the additional RNA-seq-based KD/rescue experiments we performed strongly suggest that (i) the observed effects are specific to C9ORF78 and that (ii) effects on at least some exon skipping events can only be reverted by over-expression of C9ORF78^{wt}, but not by over-expression of C9ORF78^{R41A}, and are thus most likely dependent on our observed BRR2-C9ORF78 interaction.

Finally, we would like to point out that a direct involvement of C9ORF78 in regulating alternative splicing events and a role of the observed BRR2-C9ORF78 interaction in this regulation is not

only suggested by our KD/rescue experiments, but also by our observation that putative additional interactions of C9ORF78 with other spliceosomal proteins change upon disrupting or weakening the C9ORF78-BRR2 interaction by the C9ORF78 R41A exchange (please also see the additional GO analysis we now provide in new Fig. 7a). These observations also suggest credible mechanisms by which C9ORF78 could exert such roles (please refer to our revised Discussion and responses to Reviewer 3).

We describe the new RNA-seq-based KD/rescue experiments and results in the revised manuscript (line 262):

We then transfected HEK293 cells with siRNA-resistant genes encoding either C9ORF78^{wt} or C9ORF78^{R41A}, and after two hours knocked down endogenous C9ORF78 *via* siRNAs for 72 hours. RNA-seq analysis confirmed KD of endogenous C9ORF78 and over-expression of the siRNA-resistant variants to a similar extent (Supplementary Fig. 9). rMATS analysis confirmed the global changes in alternative splicing upon C9ORF78 KD as seen in the first C9ORF78 KD experiment. Significantly changed alternative 3'-ss strongly overlapped between the two KD experiments, with almost all of the overlapping targets being NAGNAG sites (28 of 33; Fig. 5a). Strikingly, we find C9ORF78 KD-induced alternative 3'-ss skipping globally reverted upon both C9ORF78^{wt} and C9ORF78^{R41A} over-expression (Fig. 5b,c), strongly arguing for a C9ORF78-specific effect.

Skipped exon events overlapped to a lower extent between the two KD experiments (Fig. 5d) and we observed only a partial rescue of C9ORF78 KD-induced changes in exon skipping events *via* the siRNA-resistant variants (Fig. 5e). Nonetheless, 376 exon skipping events were significantly altered in both KD datasets (Δ percent spliced-in [PSI] > 0.1; $p < 0.05$), 49 of which were significantly reverted only by over-production of C9ORF78^{wt} (Fig. 5f), including skipping of PTBP2 exon 10 (Fig. 5g), indicating a regulatory mechanism that depends on the observed BRR2-C9ORF78 interaction. Together, these findings confirm that the observed alternative splicing changes upon C9ORF78 KD are indeed specific and suggest different mechanisms of splicing regulation, as C9ORF78-regulated alternative 3'-ss appear to be less dependent on the BRR2-C9ORF78 interaction than C9ORF78-regulated cassette exons.

The new results are now presented in new Supplementary Fig. 9 and new Fig. 5: New Supplementary Fig. 9:

New Fig. 5:

We also amended the Discussion section accordingly (line 381):

C9ORF78 KD elicited changes in many exon skipping events, a significant number of which are dependent on the BRR2-C9ORF78 interaction. Exon skipping is thought to be decided before the C complex stage, providing additional indirect evidence that C9ORF78 is already present at an earlier stage. We also observed a highly reproducible effect of C9ORF78 KD on alternative usage of NAGNAG 3'-ss, with C9ORF78 strongly favoring usage of the upstream 3'-ss, for which differential rescue experiments indicated C9ORF78 specificity but failed to support a dependence on the observed BRR2-C9ORF78 interaction. While these data suggest that different C9ORF78-splicing factor interactions play a predominant role for the regulation of cassette exons and of alternative 3'-ss, assay limitations may also have prevented the detection of subtle effects of the BRR2-C9ORF78 interaction on some alternative splicing events. E.g., the single residue exchange in C9ORF78^{R41A} is sufficient to destabilize the binary interaction with BRR2 *in vitro*, but C9ORF78 interaction with other spliceosomal factors and

over-expression of the siRNA-resistant C9ORF78 variants may have obscured differences in rescue efficiencies between C9ORF78^{wt} and C9ORF78^{R41A}.

Minor points

1. Figure 2a C9ORF78 labelling – there are 2 alpha 1 and no alpha 3 present.

Sorry for this mistake, corrected.

2. “We exchanged C9ORF78 F8 and R41 individually for alanine residues, and tested BRR2 binding of the C9ORF78 variants via analytical SEC. While C9ORF78F8A showed reduced binding to BRR2 in analytical SEC, BRR2 binding by C9ORF78R41A in vitro was completely abolished”

C9ORF78-F8A is not present in Figure 3b, perhaps the authors have forgotten to add the panel with WB analysis of a SEC experiment for the C9ORF78F8A mutant.

We have now included the C9ORF78^{F8A} SEC experiment in new Fig. 3b:

3. Figure 4c. The first 2 panel appear to be only depicting statistically significant splicing events, while the NAGNAG panel appears to show all events. Perhaps the authors could consider showing only statistically significant NAGNAG events similarly to the previous 2 panels of the figure.

Thank you for the suggestion. As the vast majority of regulated 3'-ss events are NAGNAG events, we decided to highlight by color all NAGNAG events in the panel showing the regulated 3'-ss events (new Fig. 4c, right):

4. Figure 5a. Labelling of y axis. It would be beneficial to explain in the text or figure legend what PSI is and how it is calculated.

This is now new Fig. 4d. We have included an explanation of PSI and how it was calculated in the figure legends of new Fig. 4 (for gel-based experiments) and new Fig. 5 (for sequencing-based experiments):

PSI, percent spliced-in (gel analysis), ratio of the quantified band representing exon inclusion and the sum of the quantified bands representing exon inclusion and exon skipping.

PSI, percent spliced-in (RNA-seq analysis), ratio of the quantified junction reads representing exon inclusion and the sum of the quantified junction reads representing exon inclusion and exon skipping.

5. Figure 5b. It would be beneficial to explain in the text or figure legend how the fold change was calculated.

This experiment has been replaced by the new RNA-seq analyses upon endogenous C9ORF78 KD and rescue with siRNA-resistant C9ORF78^{wt} or C9ORF78^{R41A} (shown in new Fig. 5; please see above).

6. Figure 6a. The authors could look at their FLASH data for sites of crosslink between C9ORF78 and U5 snRNA. With FLASH I would expect that an RNA-protein crosslink site would manifest itself as a tight cluster of deletions or point mutations in cDNA sequences. If it is possible to elucidate the exact crosslink site, it could significantly contribute to the structural data and perhaps could allow further speculations on the location of the part of C9ORF78 that is not engaged in interaction with BRR2.

Again, thanks for the suggestion. We indeed observe gaps in the U5 snRNA sequencing reads that suggest the C9ORF78 cross-linking site. They map to U5 snRNA positions 69-73, *i.e.*, to internal loop 1 of U5 snRNA. This observation is now shown in new Fig. 6b:

Furthermore, we now describe in the revised text (line 297):

A cluster of gaps in the U5 snRNA sequencing reads suggested cross-links of C9ORF78 to U5 snRNA residues 69-73, which form internal loop 1 (IL1) at the base of the extended U5 5'-stem-loop (Fig. 6b).

We also highlighted the identified cross-link site in new Fig. 8a, in which we now present the locations of putative C9ORF78 interactors in the spliceosomal B^{act} complex (please also see our replies to comments by Reviewer 3):

7. Comments on the C9ORF78 interactome.

The authors show that spliceosomal proteins are enriched in both C9ORF78-WT and C9ORF78-R41A. It would be good to see a GO term analysis, considering the manuscript reports that C9ORF78 IP results in enrichment over control for 2411 proteins. This is perhaps 1/5th of the proteome, expressed in HEK cells. In fact, the very limited GO analysis that I performed on the manuscripts data does reveal a strong enrichment for GO CC “spliceosome”, which helps interpretation of the data, as it shows that the IP has been specific.

We thank the reviewer for this suggestion, which we have implemented. Just to clarify - there were 2,411 proteins identified and quantified in all experiments combined, but not all of them were enriched over the control. Many of these proteins represent background that was not removed completely from the beads by washing. This is very common in MS-based IP analyses because of the high sensitivity. The control experiment is therefore essential to separate this non-specific background from specifically enriched proteins. To extract the specifically enriched proteins, it is necessary to define a minimum level of enrichment. In label-free quantitation at least a 2-fold enrichment (over control) should be used as a threshold. By filtering the protein list for “log2 fold change > 1” and a significant t-test result ($q < 0.05$), we ended up with 560 enriched proteins in C9ORF78^{wt} Flag-IP vs. the control, and 809 enriched proteins in C9ORF78^{R41A} Flag-IP vs. the control. This is now clarified in the revised text (line 309):

Filtering for a two-fold enrichment over the control (log2-fold change > 1) and a significant t-test result ($q < 0.05$) yielded 560 enriched proteins in the C9ORF78^{wt} Flag-IP and 809 enriched proteins in the C9ORF78^{R41A} Flag-IP. We subjected proteins with a log2-fold enrichment > 3 in either of the Flag-IPs to a GO analysis, which indicated “U5 snRNP” as the most enriched GO term (Fig. 7a).

As suggested, we have included a GO-term analysis in new Fig. 7a:

To that end, I would suggest that in addition to figure 6b the chapter “C9ORF78 interacts with additional spliceosomal proteins” could benefit from a figure that would depict the proteins, belonging to two spliceosomal complexes B and C. Those proteins should be colour coded to depict 2 key parameters, discussed in the chapter: whether a given protein was detected as enriched in the IP and if so, whether its enrichment is higher or lower in C9ORF78-WT compared to C9ORF78-R41A. I am of the opinion that this would be a great visual help for the chapter. Perhaps the authors could consider including a figure like that in the manuscript.

Again, thanks for the suggestion. We now provide a new Fig. 7b that includes all spliceosomal proteins from all complexes/stages for which our experiments show any interaction with C9ORF78 (wt or R41A; these include spliceosomal B and C complex proteins as suggested).

As a reference list of proteins, we took proteins observed in cryoEM structures of the respective spliceosomal complexes/stages, as listed in Kastner *et al.* (2019) *Cold Spring Harb Perspect Biol* **11**, a032417 (PMID: 30765414). The figure also lists all proteins of the various complexes/stages that were not enriched, as suggested. Enrichments *via* C9ORF78^{wt} and C9ORF78^{R41A} are shown by dark and light orange bars, respectively. We have experimented with more complex color coding, but found it to be confusing. For each complex/stage we now show enriched proteins in the following order:

1. Proteins enriched more by C9ORF78^{wt} than by C9ORF78^{R41A} (ordered from most to least enriched by C9ORF78^{wt})
2. Proteins enriched more by C9ORF78^{R41A} than by C9ORF78^{wt} (ordered from most to least enriched by C9ORF78^{R41A})
3. Proteins only enriched by C9ORF78^{wt} (ordered from most to least enriched)
4. Proteins only enriched by C9ORF78^{R41A} (ordered from most to least enriched)

New Fig. 7b:

Reviewer #2

The manuscript by Bergfort et al reports an unstructured protein C9ORF78 tightly interacts with the key spliceosomal RNA helicase BRR2 through a series of evidence, including yeast two-hybrid screen, in vitro protein-protein interaction, affinity purification and mass spectrometry, and cryoEM structures. They find that C9ORF78 and another spliceosomal protein FBP21 interact with the C-terminal cassette of BRR2 in a mutually exclusive manner using both structural information and biochemical competition assay. RNAi of C9ORF78 leads to alternative splicing changes including a

substantial usage of alternative 3'SSs. This manuscript provides insightful information in understanding the function of a flexibly or dynamically bound spliceosomal protein during the process of spliceosome assembly and catalysis. In general, their findings are convincing and interesting, and the manuscript is well written.

We thank the reviewer for the very positive general evaluation, in particular for considering our findings convincing and interesting and the manuscript to be well-written.

Below are several questions and concerns:

1. Typo: In Figures 1-3, several places of C9ORF78 are shown in "C9ORF8"; In Figures 2a and 3a, there are two α 1s labelled for the structure of C9ORF78, of which one should be " α 3"; In Figure 7, consistent with its legends, the forest green component marked as "PRP22" should be "PRPF22".

Sorry for these mistakes and thank you for catching them, all corrected.

2. C9ORF78 was observed in the C complex and the binding of C9ORF78 with BRR2 was also described in *S. pombe*. In Figures 1a & 1b and later, overexpression of GST- or Flag-tagged C9ORF78 are used for SEC and IP experiments, indicating that the interaction between C9ORF78 and BRR2 might be not strong in HEK293 cells. Could this be done using an endogenous normally expressed C9ORF78, either through an C9ORF78-specific antibody or by a CRSIPR-Cas9 knock-in tag system?

While we agree with the reviewer that pull-down experiments with over-expressed, tagged proteins do not appropriately reflect the strength of an interaction in cells, we also report SEC analyses in various configurations. Due to the extended times that complexes need to persist while being separated, SEC represents a rather stringent test for stable complexes. We also provide evidence *via* competitive interaction tests that C9ORF78 binds BRR2 stronger than the known BRR2 interactor, FBP21. Together with previous observations listed by the reviewer, we think that these findings corroborate our suggestion based on pull-down experiments that the interaction can also ensue in cells. We have slightly modified our corresponding statement in the revised text (line 123):

We also observed co-immunoprecipitation (co-IP) of BRR2 *via* Flag-C9ORF78 in HEK293 cells (Fig. 1b).

3. In Figure 4a, this should be RT-qPCR, not qPCR, detection of mRNAs, better to present this with an additional agarose gel.

We corrected the description. We report results from RT-qPCR, as this assay is more sensitive than agarose gel quantification. However, we have analyzed RT-qPCR products *via* agarose gels, which we are presenting below to document that we obtain single bands for C9ORF78 and GAPDH, respectively, and that the expression level of C9ORF78 is reduced upon siRNA knock-down.

4. In Figure 4b, SE (skipped exon) is the dominant feature of AS events when the C9ORF78 is KD. I am curious, what is the feature of those exons? For those increased inclusion of exons (up-regulated), do they have weaker 3'SSs; vice versa, for those decreased inclusion of exons (down-regulated), do they have stronger 3'SSs? Therefore, I would like to suggest strength analyses (scores) of both the 3'SSs and 5'SSs of those SE events.

We thank the reviewer for this suggestion and now report this information in the revised text (line 228):

Further analysis showed that C9ORF78 KD-induced exon skipping is associated with short exons, while exons included upon C9ORF78 KD exhibited an increased average length. Additionally, exons whose inclusion changed upon C9ORF78 KD showed weaker 5'-ss but average-strength 3'-ss, independent of the direction of regulation (Supplementary Fig. 6).

The analysis is now also presented in new Supplementary Fig. 6:

5. In Figure 4c, the negative value of deltaPSI are not presented, this is confusing of which sample vs which sample. In addition, RMATS should be rMATS.

Sorry for this, we have corrected both. Here the corrected Fig. 4c:

6. The primes in “3’ or 5’-splice site” are incorrect, should be 3' or 5'.

Sorry again, corrected throughout.

Reviewer #3

In this manuscript the authors present a rather comprehensive analysis of the interactions of the human splicing factor C9ORF78 with the spliceosome and particularly with the helicase Brr2, revealing a previously unappreciated role for this factor during catalysis in modulating 3'SS selection. The manuscript suggests a molecular mechanism that can explain how Brr2 may act during the catalytic stage. I generally find the authors' data compelling, of timely interest to the field, and potentially more broadly, while the proposed mechanistic models are mostly supported by the presented experiments. However, there are a few points that the authors should try to address prior to publication.

We thank the reviewer for this very positive overall evaluation, specifically for considering our data to be compelling and interesting.

While the cryo-EM analysis appears sound and expertly performed, the authors do not provide any clear figure showing their fit of the C9ORF78 model into their determined EM map. Given the claimed high resolution, it is critical for the authors to show this data as a figure in the paper, especially for the critical parts where C9ORF78 interacts with Brr2. The mutational data in Fig. 3 does support the proposed modelling, but still it is impossible to fully judge the quality of the map and the model fit without this data, especially as the presented local resolution in Sup.Fig. 2e shows significant variation in local resolution along the proposed C9ORF78 path. Certainly the map presented in Fig. 1c is contoured at an RMSD that makes it hard to judge the presence of high resolution features.

We thank the reviewer for pointing this out and now provide an additional Supplementary Figure (new Supplementary Fig. 4) documenting the quality of the cryoEM reconstructions:

Although the in vitro assays show a modest effect of C9ORF78 on Brr2 helicase activity for U4/U6, I think it is premature to argue that C9ORF78 does not act through modulation of Brr2 helicase activity, as the authors do on p.8. The assays used are not the native situation in the spliceosome and C9ORF78 binds at the U2/U6 stage of splicing, while Brr2 has been implicated in spliceosome disassembly. Thus, one could easily imagine that C9ORF78 may act at the disassembly stage and that its role in 3'SS selection could be coupled to a role in modulating Brr2 activity during disassembly. I am aware that this role for Brr2 is a matter of contention in the field but I think the authors should be more cautious with their statements here, as their data cannot exclude a role for C9ORF78 in modulating Brr2 during this later stage.

We thank the reviewer for the insightful comments. We agree and have adjusted the revised manuscript accordingly. However, we restricted the description to pointing out the possibility that C9ORF78-mediated regulation of BRR2 helicase activity may play a role, but in the absence of clear evidence refrained from speculating about a possible stage/events for which this may be important.

We changed the headline of the corresponding Results section to:

C9ORF78 moderately down-regulates BRR2 helicase activity

We left out the statement from the Results that the findings argue “against a modulation of BRR2 helicase activity constituting a major C9ORF78 function”.

We included a short section on this aspect in the revised Discussion (line 345):

The BRR2-modulatory activity of C9ORF78 we report here is weaker, and C9ORF78 seems to be associated with the spliceosome only at stages when BRR2 has already unwound U4/U6.

However, we presently cannot exclude that C9ORF78-dependent regulation of BRR2 helicase activity may play a role during other stages of splicing.

The observations regarding competition between FBP21 and C9ORF78 for Brr2 binding are strongly supported by the data in Fig. 2d. Nonetheless, it is unclear why the authors chose to use Brr2HR complexes rather than complexes that also contained the Jab1 domain of Prp8, given that the Jab1 domain remains bound to Brr2 from the B complex onwards and that the FBP21 interaction is observed in the B complex structure in the context of Brr2 being bound to Jab1. I think the authors could and likely should strengthen their argument here by performing the competition experiment with Brr2HR/Jab1/C9ORF78 complexes and even better do it in the proper physiological progression by asking whether C9ORF78 can compete off FBP21, which is what they propose actually happens during the splicing pathway.

We thank the reviewer for this suggestion and have performed the suggested experiments. We have pre-incubated BRR2^{HR}-PRPF8^{Jab1} with FBP21¹¹⁶⁻³⁷⁶, which gives rise to a stable BRR2^{HR}-PRPF8^{Jab1}-FBP21¹¹⁶⁻³⁷⁶ complex as revealed by SEC (new Fig. 2d). Upon adding GST-C9ORF78 to the pre-formed BRR2^{HR}-PRPF8^{Jab1}-FBP21¹¹⁶⁻³⁷⁶ complex, GST-C9ORF78 displaces FBP21¹¹⁶⁻³⁷⁶, as again revealed by SEC (new Fig. 2d). We used a longer FBP21 fragment for these experiments than employed for cryoEM, because PRPF8^{Jab1} and FBP21²⁰⁰⁻³⁷⁶ (the fragment used in cryoEM) cannot be distinguished on SDS-PAGE gels. We now describe these results in the revised manuscript (line 207) and display them in a new Fig. 2d:

To test mutually exclusive binding biochemically, we performed SEC analyses with a preformed BRR2^{HR}-PRPF8^{Jab1}-FBP21¹¹⁶⁻³⁷⁶ complex to which we added GST-C9ORF78 (Fig. 2d). We used a slightly longer FBP21 fragment in these experiments than in cryoEM studies to allow distinction from PRPF8^{Jab1} in SDS-PAGE. While FBP21¹¹⁶⁻³⁷⁶ formed a stable complex with BRR2^{HR}-PRPF8^{Jab1} (Fig. 2d, top), GST-C9ORF78 displaced FBP21¹¹⁶⁻³⁷⁶ from the BRR2^{HR}-PRPF8^{Jab1}-FBP21¹¹⁶⁻³⁷⁶ complex (Fig. 2d, bottom), indicating mutually exclusive binding and a stronger affinity of GST-C9ORF78 for BRR2^{HR}-PRPF8^{Jab1}. These findings suggest that C9ORF78 may aid in the displacement of FBP21 upon conversion of the B to the B^{act} complex, when FBP21 and other B-specific proteins are released.

Finally, while I appreciate the authors' model for C9ORF78 function during the C* stage as a compelling main mechanism of action, I think the authors need to be much more careful with their discussion of various proteins they claim are present at the C complexes stage, such as DDX23 or Prp6. These have generally only been detected as such in C complexes prepared by biochemical stalls that are not entirely clean and could have contaminating earlier and later complexes. A good example are studies that use a 3'SS mutation to capture C complexes and which are now known from cryo-EM studies to actually reach the C* and even P complex stage and then revert back to the C complex stage. Simply citing proteomic analyses of such preparations as evidence of stage-specific association is sufficient evidence for such claims. None of the single particle EM studies of C or C* complexes have identified any subpopulations containing

DDX23 for example, yet the authors routinely seem to suggest this protein could bind at the this stage. The claim that their IP of DDX23 occurs at the C complex stage is particularly problematic in this sense, especially as one could imagine much more easily how complexes that have transitioned to the B complex stage may not have fully lost DDX23 until the Bact stage, making the exchange of FBP21 for C9ORF78 an alternative point at which a transient interaction between C9ORF78 and DDX23 could have occurred. Similarly, the claim that Prp22 binds at the C complex stage should be revised, as what people have reported as binding at that stage most likely results from 3'SS mutant complexes that have reached the C* stage and then reverted back to the C complex stage without Prp22 dissociation, as was shown in a recent study on equilibrium of spliceosome conformations during catalysis in yeast. I urge the authors to be much more rigorous with their terminology when describing potential binding to various complexes.

Again, we very much appreciate these knowledgeable comments. Taking them into careful account, we have now much more judiciously discussed our findings in the revised manuscript. For the detailed revision of the text, please refer to our reply below the next point. The only aspect we want to bring up here is that we did not want to insinuate at all that DDX23 is normally present at the C complex stage, sorry if our descriptions were confusing. Instead, we only wanted to suggest an explanation for our observation that DDX23 was more enriched in our C9ORF78^{R41A} Flag-IP compared to the C9ORF78^{wt} Flag-IP, which we interpret to reflect the “liberation” of a DDX23-binding site on BRR2 in case of the C9ORF78^{R41A} variant, but not in case of C9ORF78^{wt}. However, as the reviewer rightly points out that C9ORF78 and DDX23 most likely do not occur in the same or directly neighboring spliceosomal intermediates, and as our descriptions were obviously confusing, we now completely left out the descriptions/discussions referring to DDX23.

Related to this matter, I find that the authors overlook too quickly a potential role for C9ORF78 in regulating splice site use also during the B to Bact transition. They observe significant numbers of exon skipping and mutually exclusive exon use changes in their KD experiments. It is much harder to imagine how such events could be regulated at the C or C* stage, but much easier to imagine how transfer of the 5'SS and docking of the BP helix at the active site, which occur during the B to Bact transition, or are influenced by the relative stability of these complexes, could impact these types of alternative splicing events. Indeed, the crosslinks to U5 snRNA are consistent with such a role and much more likely to reflect interactions at the Bact stage than at the C complex stage, when U5 is buried very deeply into the active site, making an interaction with a flexible part of C9ORF78 harder to imagine. Brr2-dependent association with Cwc27 also supports this idea, as in the Bact structure the flexible C9ORF78 residues proposed to interact with Prp22 in C*, could easily be imagined to interact with Cwc27 in Bact. Since C9ORF78 likely regulates the Bact transition, as the authors argue with strong experimental support, they should at least discuss the possibility of an earlier role in regulating alternative splicing at this stage through some of the other factors they observe in their IP studies.

Again, many thanks for these comments, we agree that in our original discussion we took the nominal association of C9ORF78 with the C complex stage too much at face value. We have amended and thoroughly reworked the corresponding Discussion sections, taking all remarks and suggestions by the reviewer into account (lines 355 and 381):

The mutually exclusive binding of FBP21 and C9ORF78 to BRR2 suggests that C9ORF78 might first bind to the spliceosome during the B-to-B^{act} transition, when FBP21 is released. While proteomics analyses have suggested that C9ORF78 might be associated with the C complex²⁷, the analyzed complexes had been enriched on a modified pre-mRNA lacking a 3'-ss AG dinucleotide and a 3'-exon⁵⁵. CryoEM and biochemical studies have shown that

spliceosomes assembled on such modified pre-mRNAs can progress to the C* complex state⁴⁰, that neighboring states tend to converge on the C complex state when exon ligation is inhibited⁵⁶ and that under appropriate conditions both catalytic steps of splicing can be reversed⁵⁷. Thus, factors identified *via* proteomics in nominal C complex preparations may to some extent represent contaminations from neighboring states. Presence of C9ORF78 already during the B^{act} stage is further supported by putative interactions we observe with the B^{act} proteins CWC22 and CWC27. However, as the C9ORF78-binding site of BRR2 remains unobstructed in C, C* and P complexes^{34,40,41,58,59}, and as we also identified putative C9ORF78 interactions with 1st step, C* and 2nd step factors, C9ORF78 may also remain bound after the B-to-B^{act} transition.

...
C9ORF78 KD elicited changes in many exon skipping events, a significant number of which are dependent on the BRR2-C9ORF78 interaction. Exon skipping is thought to be decided before the C complex stage, providing additional indirect evidence that C9ORF78 is already present at an earlier stage. We also observed a highly reproducible effect of C9ORF78 KD on alternative usage of NAGNAG 3'-ss, with C9ORF78 strongly favoring usage of the upstream 3'-ss, for which differential rescue experiments indicated C9ORF78 specificity but failed to support a dependence on the observed BRR2-C9ORF78 interaction. While these data suggest that different C9ORF78-splicing factor interactions play a predominant role for the regulation of cassette exons and of alternative 3'-ss, assay limitations may also have prevented the detection of subtle effects of the BRR2-C9ORF78 interaction on some alternative splicing events. E.g., the single residue exchange in C9ORF78^{R41A} is sufficient to destabilize the binary interaction with BRR2 *in vitro*, but C9ORF78 interaction with other spliceosomal factors and over-expression of the siRNA-resistant C9ORF78 variants may have obscured differences in rescue efficiencies between C9ORF78^{wt} and C9ORF78^{R41A}.

Exon skipping might be influenced by the kinetics with which two mutually exclusive splicing scenarios transition from the B *via* the B^{act} to the B* stage, and our findings suggest that C9ORF78 could modulate these transitions. Recently, additional assembly intermediates between the B and B^{act} stages have been characterized biochemically and structurally⁶⁰. These pre-B^{act} complexes contain, among others, reduced levels of the B-specific FBP21 protein, but also largely lack B^{act} proteins CWC22 and CWC27 and the step 1 factor GPKOW. Given our observations that C9ORF78 can displace FBP21 from BRR2 and could also contact CWC22, CWC27 and GPKOW, presence of C9ORF78 might modulate the kinetics of B-to-B^{act} conversion by driving displacement of FBP21 and helping recruitment of B^{act} proteins and GPKOW. Notably, the multi-step B-to-B^{act} transition is also accompanied by a stepwise repositioning of BRR2⁶⁰, which might likewise be influenced by C9ORF78 that putatively links BRR2 to other components according to our data. Moreover, a large-scale cryoEM analysis has revealed that the human B^{act} complex can adopt at least eight major conformations, which could be arranged along a trajectory towards catalytic activation due to the degree of their similarity to a later intermediate⁶¹. Such a situation most likely also applies to other splicing stages, and it has been suggested that any additional incoming factor will alter the conformational space available to the respective spliceosomal intermediate⁶¹. Based on a structural superposition, it is easily conceivable that in B^{act} the intrinsically unstructured C9ORF78 could bridge between BRR2, CWC22/CWC27 and U5 IL1 (Fig. 8a) This presumed cross-strutting of several B^{act} elements would most likely significantly alter the conformational space available to B^{act}. C9ORF78 might thereby again alter the kinetics of B-to-B^{act} conversion and/or influence the tendency of the B^{act} complex to adopt a conformation conducive to PRPF2 remodeling.

Alternative NAGNAG splice site choice was suggested to take place during the second step of the splicing reaction⁴⁸, indirectly supporting our notion of a continued presence of C9ORF78 at post-B^{act} stages. Furthermore, our observed interaction of C9ORF78 with the second step factor, PRPF22, suggests that C9ORF78 remains associated also with the C* complex. Comparison of our BRR2^{HR}-PRPF22-C9ORF78 structure with the structure of a human C* complex revealed that the C-terminal 231 residues of C9ORF78 could easily reach and directly contact PRPF22, which resides in immediate vicinity of BRR2 in the C* complex, as well as CWC22 that is still present at the C* stage (Fig. 8b). PRPF22 has been shown to be involved

in 3'-ss selection and exon ligation in yeast^{62,63}. Thus, one possible mechanism for C9ORF78 to regulate alternative 3'-ss usage may be direct C9ORF78-PRPF22 interactions that affect PRPF22 motor activity, which is thought to reposition alternative 3'-ss in the spliceosome's active site from a distance^{4,63}. This interpretation is consistent with our observation that presence of C9ORF78 leads to preferential use of upstream alternative 3'-ss.

New Fig. 8:

Reviewers' Comments:

Reviewer #1:

Remarks to the Author:

First of all, I wanted to say that the authors have done a remarkable job addressing the few points I raised in my original review. I believe the new data obtained by the authors has improved an already good manuscript. As with every good data, it raises more questions than it answers, which is to be expected. Furthermore the data is well presented and adequately discussed.

In addition to revealing the relatively small pool of transcripts, sensitive to the integrity of BRR2-C9ORF78 interaction, the RNA-seq results clearly hint at an exciting possibility that C9ORF78 may influence splicing outcomes via interactions with spliceosomal proteins other than BRR2.

I believe, that the manuscript in its current version contains a wealth of essential data which would be invaluable for scientists attempting to further investigate the role of C9ORF78 in splicing and beyond. I am of the opinion that the manuscript meets the expected publication standards and I raise no further issues with it.

Reviewer #2:

Remarks to the Author:

No further questions.

Reviewer #3:

Remarks to the Author:

In this revised manuscript the authors provide a compelling and timely analysis of the role of C9ORF78 in modulating splice site usage by the human spliceosome, through direct interactions with the core spliceosomal ATPase Brr2. Altogether, the structural, biochemical, and sequencing data provide strong support for the idea that C9ORF78 is recruited to the spliceosome during the pre-catalytic phase, may modulate spliceosome activation, and likely remains bound during the catalytic stage, when it influences selection of the 3'SS.

I think the revised manuscript has a better flow and the authors have presented a better and more nuanced discussion of their data.

Overall, I am satisfied that my previous concerns, as well as the potential concerns of the other reviewers, have been addressed in the revised manuscript, including by several new experiments, as requested.

I think the manuscript elucidates several functions for C9ORF78 and provides a significant advance in understanding the indirect roles played by Brr2 during later stages of splicing.

Thus I strongly recommend publication in its present form.